# Learning in Markov Games with Adaptive Adversaries: Policy Regret, Fundamental Barriers, and Efficient Algorithms

**Thanh Nguyen-Tang**
Department of Computer Science
Johns Hopkins University
Baltimore, MD 21218
nguyent@cs.jhu.edu

**Raman Arora**
Department of Computer Science
Johns Hopkins University
Baltimore, MD 21218
arora@cs.jhu.edu

## Abstract

We study learning in a dynamically evolving environment modeled as a Markov game between a learner and a strategic opponent that can adapt to the learner's strategies. While most existing works in Markov games focus on external regret as the learning objective, external regret becomes inadequate when the adversaries are adaptive. In this work, we focus on *policy regret* – a counterfactual notion that aims to compete with the return that would have been attained if the learner had followed the best fixed sequence of policy, in hindsight. We show that if the opponent has unbounded memory or if it is non-stationary, then sample-efficient learning is not possible. For memory-bounded and stationary, we show that learning is still statistically hard if the set of feasible strategies for the learner is exponentially large. To guarantee learnability, we introduce a new notion of *consistent* adaptive adversaries, wherein, the adversary responds similarly to similar strategies of the learner. We provide algorithms that achieve $\sqrt{T}$ policy regret against memory-bounded, stationary, and consistent adversaries.

## 1 Introduction

Recent years have witnessed tremendous advances in reinforcement learning for various challenging domains in AI, from the game of Go [Silver et al., 2016, 2017, 2018], real-time strategy games such as StarCraft II [Vinyals et al., 2019] and Dota [Berner et al., 2019], autonomous driving [Shalev-Shwartz et al., 2016], to socially complex games such as hide-and-seek [Baker et al., 2019], capture-the-flag [Jaderberg et al., 2019], and highly tactical games such as poker game Texas hold' em [Moravčík et al., 2017, Brown and Sandholm, 2018]. Notably, most challenging RL applications can be systematically framed as multi-agent reinforcement learning (MARL) wherein multiple strategic agents learn to act in a shared environment [Yang and Wang, 2020, Zhang et al., 2021].

Despite the empirical successes, the theoretical foundations of MARL are underdeveloped, especially in settings where the learner faces *adaptive* opponents who can strategically adapt and react to the learner's policies. Consider for example the optimal taxation problem in the AI economist [Zheng et al., 2020], a game that simulates dynamic economies that involve multiple actors (e.g., the government and its citizens) who strategically contribute to the game dynamics. The government agent learns to set a tax rate that optimizes for the economic equality and productivity of its citizens, whereas the citizens who perhaps have their own interests, respond adaptively to tax policies of the government agent (e.g., relocating to states that offer generous tax rates). Such adaptive behavior of participating agents is a crucial component in other applications as well, e.g., mechanism design [Conitzer and Sandholm, 2002, Balcan et al., 2005], optimal auctions [Cole and Roughgarden, 2014, Dütting et al., 2019].

38th Conference on Neural Information Processing Systems (NeurIPS 2024).

| Opponent's Adaptive Behavior | Policy Regret |
|---|---|
| Unbounded memory | $\Omega(T)$ |
| $m$-memory bounded ($m \geq 0$) | $\Omega(\sqrt{T(SA)^H})$ |
| $m$-memory bounded + stationary ($m \geq 1$) | $\Omega(\min\{T, A^{HS}\})$ |
| 1-memory bounded + stationary + consistent | $\tilde{\mathcal{O}}(H^3 S^2 AB + \sqrt{H^5 SA^2 BT})$ |
| $m$-memory bounded + stationary + consistent | $\tilde{\mathcal{O}}\left((m-1)H^2 SAB + \sqrt{H^3 SAB}(SAB(H + \sqrt{S}) + H^2)\sqrt{\frac{T}{d^*}}\right)$ |

Table 1: Summary of main results for learning against adaptive adversaries. Learner's policy set is all deterministic Markov policies. $m = 0$ + stationary corresponds to standard single-agent MDPs.

The question of learning against adaptive opponents has been mostly studied under the framework of external regret, wherein the agent is required to compete with the best fixed policy in hindsight [Liu et al., 2022]. However, external regret is not adequate to study adaptive opponents as it does not take into account the counterfactual response of the opponents. This motivates us to study MARL using the framework of *policy regret* [Arora et al., 2012], a counterfactual notion that aims to compete with the return that would have been attained if the agent had followed the best fixed sequence of policy in hindsight. Even though policy regret is now a standard notion to study adaptive adversaries and has been extensively studied in online (bandit) learning [Merhav et al., 2002, Arora et al., 2012, Malik et al., 2022] and repeated games [Arora et al., 2018], it has not received much attention in a multiagent reinforcement learning setting. In this paper, we aim to fill in this gap. We consider two-player Markov games (MGs) [Shapley, 1953, Littman, 1994] as a model for MARL, wherein one agent (the learner) learns to act against an adaptive opponent. We provide a series of negative and positive results for policy regret minimization in Markov games, highlighting the fundamental limits of learning and showcasing key principles underpinning the design of efficient learning algorithms against adaptive adversaries.

**Fundamental barriers.** We first show that any learner must incur a linear policy regret against an adaptive opponent who can adapt and remember the learner's past policies (Theorem 1). When the opponent has a bounded memory span, any learner must require an exponential number of samples $\Omega((SA)^H/\epsilon^2)$ to obtain an $\epsilon$-suboptimal policy regret, even with the weakest form of memory wherein the opponent is oblivious (Theorem 2). When the memory-bounded opponent's response is stationary, i.e., the response function does not vary with episodes, learning is still statistically hard when the learner's policy set is exponentially large, as in this case the policy regret necessarily scales polynomially with the cardinality of the learner's policy set (Theorem 3).

**Efficient algorithms.** Motivated by these statistical hardness results, we consider a structural condition on the response of the opponents, which we refer to as consistent behavior, wherein the opponent responds similarly to similar sequences of policies (Definition 5). We propose two algorithms OPO-OMLE (Algorithm 1) and APE-OVE (Algorithm 3) that obtain $\sqrt{T}$ policy regret against $m$-memory bounded, stationary, and consistent adversaries, for $m = 1$ and $m \geq 1$, respectively.

- **For memory length $m = 1$:** We show that OPO-OMLE obtains a policy regret upper bound of $\tilde{\mathcal{O}}(H^3 S^2 AB + \sqrt{H^5 SA^2 BT})$, when the learner's policy set is the set of all deterministic Markov policies, where $H$ is the episode length, $S$ is the number of states, $A$ and $B$ are the numbers of actions for the learner and the opponent, respectively, and $T$ is the number of episodes.

- **For general memory length $m \geq 1$:** We show that APE-OVE obtains a policy regret upper bound of $\tilde{\mathcal{O}}\left((m-1)H^2 SAB + \sqrt{H^3 SAB}(SAB(H + \sqrt{S}) + H^2)\sqrt{\frac{T}{d^*}}\right)$, where $d^*$ is an instance-dependent quantity that features the minimum positive visitation probability.

We provide a summary of our main results in Table 1.

## 2 Related work

**Learning in Markov games.** Learning problems in Markov games have been studied extensively in the MARL literature. Most existing works focus on learning Nash equilibria either with known dynamics or infinite data [Littman, 1994, Hu and Wellman, 2003, Hansen et al., 2013, Wei et al., 2020], or otherwise in a self-play setting wherein we control all the players [Wei et al., 2017, Bai et al., 2020, Bai and Jin, 2020, Xie et al., 2020, Liu et al., 2021], or in an online setting wherein we

control one player to learn against other potentially adversarial players [Brafman and Tennenholtz, 2002, Wei et al., 2020, Tian et al., 2021, Jin et al., 2022]. Other related work focuses on exploiting sub-optimal opponents via no-external regret learning [Liu et al., 2022] and studying Stackelberg equilibria in two-player general-sum turn-based MGs, wherein only one player is allowed to take actions in each state [Ramponi and Restelli, 2022].

**Policy regret in online learning settings.** Policy regret minimization has been studied mostly in online (bandit) learning problems. It was first studied in a full information setting [Merhav et al., 2002] and extended to the bandit setting and more powerful competitor classes using swap regret and $\Phi$-regret [Arora et al., 2012]. A lower bound of $T^{2/3}$ on policy regret in a bandit setting was provided by Dekel et al. [2014] and was later extended to action space with metric [Koren et al., 2017a,b]. A long line of works studies (complete) policy regret in "tallying" bandits, wherein an action's loss is a function of the number of the action's pulls in the previous $m$ rounds [Heidari et al., 2016, Levine et al., 2017, Seznec et al., 2019, Lindner et al., 2021, Awasthi et al., 2022, Malik et al., 2022, 2023].

Beyond online (bandit) learning, policy regret has been studied in several more challenging settings. In Arora et al. [2018] authors study the notion of policy equilibrium in repeated games (Markov games with $H = S = 1$) when agents follow no-policy regret algorithms. A more complete characterization of the learnability in online learning with dynamics, where the loss function additionally depends on time-evolving states, was given in Bhatia and Sridharan [2020]. Finally, in Dinh et al. [2023], authors study policy regret in online MDP, where an adversary who follows a no-external regret algorithm generates the loss functions, which effectively alleviates policy regret minimization to the standard external regret minimization in online MDPs.

## 3 Problem setup

**Markov games.** In this paper, we use the framework of Markov Games to study an interactive multi-agent decision-making and learning environment [Shapley, 1953]. Markov games extend Markov decision processes (MDPs) to multiplayer scenarios, where each agent's action affects not only the environment but also the subsequent state of the game and the actions of other agents. Formally, a standard two-player Markov Game (MG) is specified by a tuple $M = (\mathcal{S}, \mathcal{A}, \mathcal{B}, H, P, r)$. Here, $\mathcal{S}$ denotes the state space with cardinality $|\mathcal{S}| = S$, $\mathcal{A}$ is the action space of the first player (called *learner*) with cardinality $|\mathcal{A}| = A$, $\mathcal{B}$ is the action space of the second player (referred to as an *opponent* or an *adversary*) with cardinality $|\mathcal{B}| = B$, $H \in \mathbb{N}$ is the time horizon for each game. $P = \{P_1, \ldots, P_H\}$ are the transition kernels with each $P_h : \mathcal{S} \times \mathcal{A} \times \mathcal{B} \to \Delta(S)$ specifying the probability of transitioning to the next state given the current state, learner's action, and adversary's action ($\Delta(\mathcal{S})$ denotes the set of all probability distributions over $\mathcal{S}$). Finally, $r = \{r_1, \ldots, r_H\}$ are the (expected) reward functions with each $r_h : \mathcal{S} \times \mathcal{A} \times \mathcal{B} \to [0, 1]$. For simplicity, we assume the learner knows the reward function.[1]

Each episode begins in a fixed initial state $s_1$. At step $h \in [H]$, the learner observes the state $s_h$ and picks her action $a_h \in \mathcal{A}$ while the opponent/adversary picks an action $b_h \in \mathcal{B}$. As a result, the learner observes $b_h$, receives reward $r_h(s_h, a_h, b_h)$ and the environment transitions to $s_{h+1} \sim P_h(\cdot|s_h, a_h, b_h)$. The episode terminates after $H$ steps.

**Policies and value functions.** A learner's policy (also referred to as strategy) is any tuple $\pi = \{\pi_h\}_{h \in [H]}$ where $\pi_h : (\mathcal{S} \times \mathcal{A})^{h-1} \times \mathcal{S} \to \Delta(\mathcal{A})$. A policy $\pi = \{\pi_h\}_{h \in [H]}$ is said be Markovian if for every $h \in [H], \pi_h : \mathcal{S} \to \Delta(\mathcal{A})$. Similarly, an adversary's policy is any tuple $\mu = \{\mu_h\}_{h \in [H]}$ where $\mu_h : (\mathcal{S} \times \mathcal{B})^{h-1} \times \mathcal{S} \to \Delta(\mathcal{B})$. $\mu$ is said to be Markovian if for every $h, \mu_h : \mathcal{S} \to \Delta(\mathcal{B})$. For simplicity, we will focus only on Markov policies for both the learner and the adversary in this paper. Let $\Pi$ (respectively, $\Psi$) be the set of all feasible policies of the learner (respectively, the adversary). The value of a policy tuple $(\pi, \mu) \in \Pi \times \Psi$ at step $h$ in state $s$, denoted by $V_h^{\pi,\mu}(s)$ is the expected accumulated reward starting in state $s$ from step $h$, if the learner and the adversary follow $\pi$ and $\mu$ respectively, i.e., $V_h^{\pi,\mu}(s) := \mathbb{E}_{\pi,\mu}[\sum_{l=h}^{H} r_l(s_l, a_l, b_l)|s_h = s]$, where the expectation is with respect to the trajectory $(s_1, a_1, b_1, r_1, \ldots, s_H, a_H, b_H, r_H)$ distributed according to $P, \pi$, and $\mu$. We also denote the action-value function $Q_h^{\pi,\mu}(s, a, b) := \mathbb{E}_{\pi,\mu}[\sum_{l=h}^{H} r_l(s_l, a_l, b_l)|(s_h, a_h, b_h) = (s, a, b)]$.

---

[1]Our results immediately generalize to unknown reward functions, as learning the transitions is more difficult than learning the reward functions in tabular MGs.

Given a $V : \mathcal{S} \to \mathbb{R}$, we write $P_h V(s, a, b) := \mathbb{E}_{s' \sim P_h(\cdot|s,a,b)}[V(s')]$. For any $u : \mathcal{S} \to \Delta(\mathcal{A})$, $v : \mathcal{S} \to \Delta(\mathcal{B})$, $Q : \mathcal{S} \times \mathcal{A} \times \mathcal{B} \to \mathbb{R}$, denote $Q(s, u, v) := \mathbb{E}_{a \sim u(\cdot|s), b \sim v(\cdot|s)}[Q(s, a, b)]$ for any $s \in \mathcal{S}$.

**Adaptive adversaries.** We allow the adversary to be *adaptive*, i.e., the adversary can choose their policy in episode $t$ based on the learner's policies on episodes $1, \dots, t$. We assume that the adversary is deterministic and has unlimited computational power, i.e., the adversary can plan, in advance, using as much computation as needed, as to how they would react in each episode to any sequence of policies. Formally, the adversary defines in advance a sequence of deterministic functions $\{f_t\}_{t \in \mathbb{N}^*}$, where $f_t : \Pi^t \to \Psi$. The input to each response function $f_t$ is an entire history of the learner's policies, including her policy in episode $t$. Therefore, if the learner follows policies $\pi^1, \dots, \pi^t$, the adversary responds with policy $f_t(\pi^1, \dots, \pi^t) \in \Psi$ in episode $t$. Since the response function $f_t$ depends on the learner's policy at round $t$, our setup is essentially a principal-follower model, akin to Stackelberg games [Letchford et al., 2009, Blum et al., 2014] and mechanism design for learning agents [Braverman et al., 2019]. In this context, the principal agent (mechanism designer or learner) publicly declares a strategy before committing to it, allowing the followers to subsequently choose their strategies based on their understanding of the principal's decisions.

We evaluate the learner's performance using the notion of **policy regret** [Merhav et al., 2002, Arora et al., 2012], which compares the return on the first $T$ episodes to the return of the best fixed sequence of policy in hindsight. Formally, the learner's policy regret after $T$ episodes is defined as

$$\mathrm{PR}(T) = \sup_{\pi \in \Pi} \sum_{t=1}^{T} V_1^{\pi, f_t([\pi]^t)}(s_1) - V_1^{\pi^t, f_t(\pi^1, \dots, \pi^t)}(s_1), \text{ where } f_t([\pi]^t) := f_t(\underbrace{\pi, \dots, \pi}_{t \text{ times}}). \quad (1)$$

Policy regret has been studied in online (bandit) learning [Merhav et al., 2002, Arora et al., 2012] and repeated games [Arora et al., 2018], yet, to the best of our knowledge, it has never been studied in Markov games. Policy regret differs from the more common definition of external regret defined as $R(T) = \sup_{\pi \in \Pi} \sum_{t=1}^{T} V_1^{\pi, f_t(\pi^1, \dots, \pi^t)}(s_1) - V_1^{\pi^t, f_t(\pi^1, \dots, \pi^t)}(s_1)$, which is used in [Liu et al., 2022]. However, external regret is inadequate for measuring the learner's performance against an adaptive adversary. Indeed, when the adversary is adaptive, the quantity $V_1^{\pi, f_t(\pi^1, \dots, \pi^t)}$ is hardly interpretable anymore – see [Arora et al., 2012] for a more detailed discussion.

As a warm-up, we show in the following example that, policy regret minimization generalizes the standard Nash equilibrium learning problem in zero-sum two-player Markov games.

**Example 3.1** (Nash equilibrium). *Consider the adversary with the following behavior: for any Markov policy $\pi$ of the learner, the adversary ignores all the learner's past policies and respond only to the current policy $\pi$ with a Markov policy $f(\pi)$ such that for all $(s, h)$, $V_h^{\pi, f(\pi)}(s) = \min_\mu V_h^{\pi, \mu}(s)$, where the minimum is taken over all the possible Markov policies for the adversary. By Filar and Vrieze [2012], such an $f(\pi)$ exists. In addtion, there also exists a Markov policy $\pi^*$ such that for all $(s, h)$, $V_h^{\pi^*, f(\pi^*)}(s) = \sup_\pi V_h^{\pi, f(\pi)}(s) = \inf_\mu \sup_\pi V_h^{\pi, \mu}(s)$. The policies $(\pi^*, f(\pi^*))$ is a Nash equilibrium [Nash, 1950] of the Markov game. For such an adversary, the policy regret becomes $PR(T) = \sum_{t=1}^{T} V_1^{\pi^*, f(\pi^*)}(s_1) - \sum_{t=1}^{T} V_1^{\pi^t, f(\pi^t)}(s_1)$. This Nash equilibrium can be computed using, e.g., the Q-ol algorithm of [Tian et al., 2021] with $\sqrt{T}$ (policy) regret.[2]*

**Additional notation.** We write $f \lesssim g$ to mean $f = \mathcal{O}(g)$. We use $c$ to represent an absolute constant that can have different values in different appearances.

## 4 Fundamental barriers for learning against adaptive adversaries

In this section, we show that achieving low policy regret in Markov games against an adaptive adversary is statistically hard when (i) the adversary has an unbounded memory (see Definition 1), or (ii) the adversary is non-stationary, or (iii) the learner's policy set is exponentially large (even if the adversary is memory-bounded and stationary).

---

[2]Q-ol algorithm solves a problem that is a bit more general than the policy regret minimization in Example 3.1 in that as long as the benchmark is the Nash value $V_1^{\pi^*, f(\pi^*)}$, regardless of the behavior of the adversary, the said rate for the policy regret is guaranteed. V-learning algorithm of Jin et al. [2021] solves a similar problem but in a self-play setting; it is not immediately clear if their rate remains in the online setting.

To begin with, we show that any learner must incur a linear policy regret in the general setting.

**Theorem 1.** *For any learner, there exists an adaptive adversary and a Markov game instance such that $PR(T) = \Omega(T)$.*

The construction in the proof of Theorem 1, shown in Appendix A.1, takes advantage of the unbounded memory of the adversary, that can remember the policy the learner takes in the first episode. This motivates us to consider memory-bounded adversaries, a situation that is quite similar to the online bandit learning setting of Arora et al. [2012].

**Definition 1** ($m$-memory bounded adversaries). *An adversary $\{f_t\}_{t \in \mathbb{N}^*}$ is said to be $m$-memory bounded for some $m \geq 0$ if for every $t$ and policy sequence $\pi^1, \ldots, \pi^t$, we have $f_t(\pi^1, \ldots, \pi^t) = f_t(\pi^{\min\{1, t-m+1\}}, \ldots, \pi^t)$.*

Is it possible to efficiently learn against memory-bounded adversaries? Unlike online bandit learning, we show that learning in Markov games is statistically hard even when the adversary is memory-bounded, even for the weakest case of memory $m = 0$ and the adversary's policy set $\Psi$ is small.

**Theorem 2.** *For any learner and any $L \in \mathbb{N}$ and $S, A, H$, there exists an oblivious adversary (i.e., $m = 0$) with the policy space $\Psi$ of cardinality at least $L$, a Markov game (with $SA + S$ states, $A$ actions for the learner, $B = 2S$ actions for the adversary) such that $PR(T) = \Omega\left(\sqrt{T(SA/L)^L}\right)$.*

Theorem 2 claims that competing even with an oblivious adversary that employs a small set of policies takes an exponential number of samples (e.g., set $S = L = H$). The construction of the lower bound follows the construction used to prove a lower bound for learning latent MDPs [Kwon et al., 2021] and a reduction of a given latent MDP into a Markov game [Liu et al., 2022]; we give complete details in Appendix A.2. The proof of Theorem 2 utilizes the fact that the sequence of response function an adversary utilizes can be completely arbitrary. It implies that we need to constrain the adversary further beyond being memory-bounded. A natural restriction we consider given the construction is to assume stationarity, i.e. consider adversaries whose response functions do not change over time.

**Definition 2** (Stationary adversaries). *An $m$-memory bounded adversary is said to be $\underline{stationary}$ if there exists an $f : \Pi^m \to \Psi$ such that for all $t$ and $\pi^1, \ldots, \pi^t$, we have $f_t(\pi^1, \overline{\ldots, \pi^t}) = f(\pi^{\min\{1, t-m+1\}}, \ldots, \pi^t)$.*

The stationary behavior is sometimes also referred to as "g-restricted" in the online learning literature–see the related discussion of Malik et al. [2022]. In the special case wherein the adversary is both stationary and oblivious (i.e., $m = 0$), the Markov game reduces to the standard single-agent MDP (and the policy regret reduces to standard regret of the MDP) – this setting has been studied in [Zhang et al., 2023]. We, therefore, only need to consider $m \geq 1$.

**Connections to Stackelberg equilibrium in general-sum Markov games.** While seemingly restrictive, policy regret minimization with $m$-memory bounded and stationary adversaries already subsumes the problem of learning Stackelberg equilibrium [Von Stackelberg, 2010] in general-sum Markov games [Ramponi and Restelli, 2022].[3] In general-sum Markov games, the adversary ("follower") aims at maximizing his own reward function given any policy of the learner ("leader"). That is, the adversary is 1-memory bounded, and the response function $f : \Pi \to \Psi$ corresponds to a function that selects the best response policy to any given policy of the learner. The benchmark $\max_{\pi \in \Pi} V_1^{\pi, f(\pi)}$ in policy regret then becomes the Stackelberg equilibrium.

Is sample-efficient learning possible against $m$-memory bounded and stationary adversaries? One can notice an immediate approach to learning against a 1-memory bounded and stationary adversaries is to simply view the problem as a $|\Pi|$-armed bandit problem and apply any state-of-the-art bandit algorithm [Audibert and Bubeck, 2009] to obtain $PR(T) = \mathcal{O}(H\sqrt{T|\Pi|})$. However, scaling polynomially with the learner's policy class is not desirable when the class is exponentially large (e.g., when the learner's policy class is the set of all deterministic policies, then $|\Pi| = \Theta(A^{HS})$). And in fact, we cannot avoid polynomial scaling with the cardinality of the learner's policy class in general.

**Theorem 3.** *For any learner with policy class $\Pi$, there exists a 1-memory bounded and stationary adversary and a Markov game with $B = \mathcal{O}(1)$ such that $PR(T) = \Omega(\min\{T, |\Pi|\})$.*

Note that the lower bound applies to $m = 1$, and, therefore, to any $m \geq 1$. Proof in Appendix A.3.

---

[3]Ramponi and Restelli [2022] consider a more restrictive setting of turn-based Markov games, wherein at each state only one player is allowed to take actions. In addition, they require the opponents to respond with only deterministic policies.

# 5 Efficient algorithms for learning against adaptive adversaries

Thus far, we have shown that learning against an adaptive adversary in Markov games is statistically hard, even when the adversary is $m$-memory bounded and stationary. The reason that stationarity is not sufficient for efficient learning (which the lower bound in Theorem 3 exploits for the construction of a hard instance) comes from the unstructured response of the adversary in the worst case. Even if the learner plays *nearly* identical sequence of policies differing only on a small number of states and steps, the adversary can essentially respond completely arbitrarily. In other words, knowing the policies that the adversary plays in response to the policies of the learner (i.e., observing the values of the response function $f$ at specific inputs) reveals zero information about the function $f$ on previously seen inputs. Thus, the learner is required to explore all the policies in $\Pi$ to be able to identify an optimal policy. This motivates us to consider an additional structural assumption on how the adversary responds to the learner's policies. We assume that the adversary is consistent in response to two similar sequences of policies of the learner. In essence, given that the learner plays two sequences of policies that agree on certain states ($s$) and steps ($h$) – then, we assume that the opponent also responds with two sequences of policies that agree on the same states and steps. We refer to this behavior as *consistent*; a formal definition follows.

**Definition 3** (Consistent adversaries). *An $m$-memory bounded and stationary adversary $f$ is said to be consistent if, for any two sequences of learner's policies $\pi^1, \ldots, \pi^m$ and $\nu^1, \ldots, \nu^m$, and any $(s, h) \in \mathcal{S} \times [H]$, if $\pi_h^i(\cdot|s) = \nu_h^i(\cdot|s), \forall i \in [m]$, then $f(\pi^1, \ldots, \pi^m)_h(\cdot|s) = f(\nu^1, \ldots, \nu^m)_h(\cdot|s)$. Otherwise, we say that the opponent's response $f$ is arbitrary.*

We argue that the definition above is natural if we are to consider opponents that are self-interested strategic agents, and not simply a malicious adversary. So, it would be in an opponent's interest to play in a somewhat consistent manner. Playing optimally after figuring out the learner's strategy would indeed require playing consistently. An opponent that plays completely arbitrary, while challenging to learn anything from, also does not improve their value function. Some remarks are in order.

**Remark 1** ($\zeta$-approximately consistent adversaries). *Our algorithms and results for consistent adversaries easily extend to $\zeta$-approximately consistent adversaries for any fixed constant $\zeta \geq 0$. An adversary $f$ is said to be $\zeta$-approximately consistent if, for any $\pi^1, \ldots, \pi^m$ and $\nu^1, \ldots, \nu^m$, and*

*any $(s, h) \in \mathcal{S} \times [H]$, if $\pi_h^i(\cdot|s) = \nu_h^i(\cdot|s), \forall i \in [m]$, then $\max_{a \in \mathcal{A}} \left| \log \frac{f(\pi^1, \ldots, \pi^m)_h(a|s)}{f(\nu^1, \ldots, \nu^m)_h(a|s)} \right| \leq \zeta$. For*

*simplicity, we stick with Definition 3 (i.e., $\zeta = 0$) to best convey our algorithmic and theoretical ideas.*

**Remark 2.** *While our notion of consistent behaviors is quite natural, it might as well be that there is a more general notion of complexity for the opponent's response function classes that fully characterizes learnability in this setting. This likely requires the definition of appropriate norms in the input policy space $\Pi^m$ and the output policy space $\Psi$, and a certain notion of predictability for the opponent's response function classes (e.g., in the spirit of Eluder dimension [Russo and Van Roy, 2013]), so that the learner can accurately estimate the opponent's response function, without trying out all possible policies. This question goes beyond the scope of our current work and is left to a future investigation.*

**Remark 3.** *To permit learnability in terms of external regret in Markov games, Liu et al. [2022] consider a policy-revealed setting, wherein the opponent reveals his current strategy to the learner at the end of each episode. No external regret is possible because the benchmark in external regret evaluates the learner's comparator policy against the same policy that the opponent reveals. For policy regret, however, knowing the opponent's strategy at the end of the episode gives the learner no advantage in general, as the counterfactual benchmark requires evaluating the learner's policies against the policy sequence that the opponent would have reacted with. Indeed, our lower bound in Theorem 3 still applies to the policy-revealed setting.*

For $m$-memory bounded, stationary and consistent adversaries, we present two algorithms, one for $m = 1$ and the other for general $m \geq 1$, with sublinear policy regret. We give special consideration to the case with $m = 1$ as it helps with the exposition of key algorithmic design principles rather simply. For simplicity, we focus on $\Pi$ being the set of all deterministic policies (i.e., $|\Pi| = \Theta(A^{HS})$). Our algorithms and upper bounds easily extend to any general $\Pi$ with polynomial log-cardinality.

**Assumption 5.1.** *The learner's policy class $\Pi$ is the set of all deterministic policies.*

A key component of our algorithms is using maximum likelihood estimation (MLE) [Geer, 2000] to estimate action distributions with which the opponent can respond. As is the convention in MLE analysis, we make a realizability assumption and use bracketing numbers to control the model class.

**Assumption 5.2.** *For any policy $\mu \in \Psi$ that the adversary employs and for all $(h, s) \in [H] \times \mathcal{S}$, assume $\mu_h(\cdot|s) \in P_\Theta := \{P_\theta \in \Delta(\mathcal{B}) : \theta \in \Theta\}$, where the set $P_\Theta$ has $\epsilon$-bracketing number $\mathcal{N}_\Theta(\epsilon)$ w.r.t. $l_1$ norm, defined as the minimum number of $\epsilon$-brackets $[l, u] := \{P_\theta \in P_\Theta : l \le P_\theta \le u\}$ with $\|l - u\|_1 \le \epsilon$, that are needed to cover $P_\Theta$.*

Intuitively, restricting the adversary to be consistent, allows the learner to predict the opponent's response from previous episodes to similar settings. The learner can collect the data from what the adversary responds to and learn his response function. Given the consistent behavior, for every $(h, s) \in [H] \times \mathcal{S}$, the number of action distributions $\mu_h(\cdot|s)$ that the adversary can respond with cannot exceed the number of possible action distributions $\pi_h(\cdot|s)$ that the learner can construct in state $s$ at step $h$. Given $\Pi$ is the set of all deterministic policies, we only need to learn $HSA$ action distributions that the adversary can respond at any state and step. We begin with the oblivious case of $m = 1$ and end up resolving the general case $m \ge 1$ after.

### 5.1 Memory of length $m = 1$

We first consider the memory length of $m = 1$ for stationary and consistent adversaries.

**Algorithm.** We propose OPO-OMLE (Algorithm 1), which represents Optimistic Policy Optimization with Optimistic Maximum Likelihood Estimation. OPO-OMLE is a variant of the optimistic value iteration algorithm of [Azar et al., 2017], wherein we build an upper confidence bound on the value function $V_1^{\pi, f(\pi)}$ for any policy $\pi$, using a bonus function and optimistic MLE [Liu et al., 2023]. The upper confidence bound is based on two levels of optimism: a bonus term $\beta$ that is based on confidence intervals on the transition kernels $P$ and the parameter version spaces $\{\Theta_{hsa}\}$ of the adversary's response at each level $(h, s, a)$. The parameter version spaces construct a set of parameters that are close to the MLE solution, up to an error $\alpha$, in terms of the log-likelihood in the observed actions taken by the adversary.

---

**Algorithm 1** Optimistic Policy Optimization with Optimistic MLE (OPO-OMLE)

1: **Input**: Bonus function $\beta : \mathbb{N} \to \mathbb{R}$, and MLE confidence parameter $\alpha$
2: **Initialize**: $\Theta_{hsa} \leftarrow \Theta, D_{hsa} \leftarrow \emptyset, N_h(s, a, b) \leftarrow 0, N_h(s, a, b, s') \leftarrow 0, \forall (h, s, a, b, s') \in \mathcal{S} \times \mathcal{A} \times \mathcal{B} \times \mathcal{S}$
3: **for** episode $t = 1, \dots, T$ **do**
4: $\quad \pi^t \in \arg\max\limits_{\pi \in \Pi} \text{DOUBLY\_OPTIMISTIC\_VALUE\_ESTIMATE}(N, \{D_i\}, \{\Theta_i\}, \pi, \beta)$
$\quad$ (Algorithm 2)
5: $\quad$ Play $\pi^t$ (the opponent responds with $f(\pi^t)$) to observe $(s_1^t, a_1^t, b_1^t, r_1^t, \dots, s_H^t, a_H^t, b_H^t, r_H^t)$
6: $\quad \forall h: N_h(s_h^t, a_h^t, b_h^t) \leftarrow N_h(s_h^t, a_h^t, b_h^t) + 1, N_h(s_h^t, a_h^t, b_h^t, s_{h+1}^t) \leftarrow N_h(s_h^t, a_h^t, b_h^t, s_{h+1}^t) + 1, D_{hs_h^t a_h^t} \leftarrow D_{hs_h^t a_h^t} \cup \{b_h^t\}$, and $\Theta_{hs_h^t a_h^t} \leftarrow \{\theta \in \Theta_{hs_h^t a_h^t} : \sum_{b \in D_{hs_h^t a_h^t}} \log P_\theta(b) \ge \max_{\theta \in \Theta_{hs_h^t a_h^t}} \sum_{b \in D_{hs_h^t a_h^t}} \log P_\theta(b) - \alpha\}$
7: **end for**
8: **Output**: $\{\pi^t\}_{t \in [T]}$

---

**Algorithm 2** DOUBLY_OPTIMISTIC_VALUE_ESTIMATE$(N, \{D_i\}, \{\Theta_i\}, \pi, \beta)$

1: **Initialize**: $\bar{V}_{H+1}^\pi = 0$
2: $\hat{P}_h(s'|s, a, b) = \frac{1}{S}$ if $N_h(s, a, b) = 0$; otherwise, $\hat{P}_h(s'|s, a, b) = N_h(s, a, b, s')/N_h(s, a, b)$
3: **for** $h = H, H-1, \dots, 1$ **do**
4: $\quad \bar{Q}_h^\pi(s, a, b) = \min\left\{[\hat{P}_h \bar{V}_{h+1}^\pi](s, a, b) + r_h(s, a, b) + \beta(N_h(s, a, b)), H - h + 1\right\}, \forall (s, a, b)$
5: $\quad \bar{V}_h^\pi(s) = \max_{\theta \in \Theta_{hs\pi_h(s)}} \bar{Q}_h^\pi(s, \pi_h, P_\theta), \forall s$ $\qquad \qquad \triangleright$ *Optimistic MLE*
6: **end for**
7: **Output**: $\bar{V}_1^\pi$

---

**Theoretical guarantee.** We now present a theoretical guarantee for OPO-OMLE.

**Theorem 4.** *In Algorithm 1, choose $\beta(t) = cH\sqrt{\frac{\iota + \log |\Pi|}{t}}$, where $\iota := \log(SABHT/\delta)$, and $\alpha = c\log(\mathcal{N}_\Theta(1/T)HSAT/\delta)$. With probability at least $1 - \delta$, we have*

$$PR(T) = \mathcal{O}\left(H^3 S^2 AB\iota \log T + H^2\sqrt{SABT(\iota + \log |\Pi|)} + H^2\sqrt{SAT\alpha}\right).$$

Theorem 4 shows that OPO-OMLE achieves $\sqrt{T}$-policy regret bounds against 1-memory bounded, stationary and consistent adversaries in Markov games. Notably, the policy regret depends only on the log-cardinality of the learner's policy class $\Pi$ and the log-bracketing number of the set of action distributions with which the adversary responds to the learner. Since $|\Pi| = A^{HS}$, the bound translates into $\text{PR}(T) = \tilde{\mathcal{O}}(H^3 S^2 AB + \sqrt{H^5 SA^2 BT})$.

Finally, comparing the lower bound of $\Omega(\min\{\sqrt{H^3 SAT}, HT\})$ for single-agent MDPs [Domingues et al., 2021], which applies to this setting, the dominating term in our upper bound (Theorem 4) is worse only by a factor of $H\sqrt{AB}$ – this is due to the need to learn the opponent's moves.[4]

## 5.2 Memory of any fixed length $m \geq 1$

We now consider the general case of stationary and consistent adversaries that have a memory of any fixed length $m \geq 1$. Note that we assume that the learner knows (an upper bound of) $m$. Playing against a 1-memory bounded adversary does not stop the learner from changing her policies often, as the adversary does not remember any policies that the learner has taken previously. However, a sublinear policy regret learner against $m$-memory bounded adversaries should switch her policies as less frequently as possible, and at most only sublinear time switches. The reason is that every policy switch will add a constant cost to policy regret, as the benchmark in the policy regret is with the best *fixed* sequence of policy. This makes the regret minimizer OPO-OMLE unable to generalize from $m = 1$ to any fixed $m$. Instead, we propose a low-switching algorithm, in which the learner learns to play *exploratory* policies repeatedly over consecutive episodes so that the switching cost is reduced. Here, as in Jin et al. [2020], exploratory policies are those with good coverage over the state space from which uniform policy evaluation can be performed to identify near-optimal policies.

**Algorithm.** We propose APE-OVE (Algorithm 3), which represents Adaptive Policy Elimination by Optimistic Value Estimation. APE-OVE generalizes the adaptive policy elimination algorithm of [Qiao et al., 2022] for MDPs to Markov games with unknown opponents. The high-level idea of our algorithm is as follows. The learner maintains a version space $\Pi^k$ of remaining high-quality policies after each epoch – which is a sequence of consecutive episodes with an appropriate length (epoch $k$ has a length of $HSAB(m - 1 + T_k)$ in APE-OVE).

- **Layerwise exploration** (Line 5 of Algorithm 3): Within each epoch, the learner performs layerwise exploration (Algorithm 4), wherein we devise high-coverage sampling policies $\pi^{khsab}$ that aim at exploring $(s, a, b)$ in step $h$ and epoch $k$, starting from the lowest layer $h = 1$ up to the highest layer $h = H$. However, some states might not be visited frequently by any policy, thus taking a large amount of exploration. They, fortunately, do not significantly affect the value functions of any policy and thus can be identified (by storing in $\mathcal{U}^k$) and removed from exploration quickly (via the truncated transition kernel estimates $\hat{P}$ obtained in Algorithm 5). Layerwise exploration requires value estimation uniformly over all policies. However, the learner does not know the adversary's response $f$. To address this, we use optimistic value estimation via the optimistic MLE in the collected data of the adversary's moves (Algorithm 6).

- **Version space refinement** (Line 6 of Algorithm 3): After the layerwise exploration, we refine the version space of policies that the learner can choose from at the next epoch using the optimistic value estimation based on the empirical transition kernels $\hat{P}^k$, the parameter version space $\Theta^k$ and the set of infrequent transition samples $\mathcal{U}^k$ given any reward function $r$. The version space is designed in such a way that the expected value for the learner to play any policy $\pi$ from the version space is guaranteed to be no worse than $\tilde{\mathcal{O}}(1/\sqrt{T_k})$ compared to the optimal, with high probability.

Note that we do not directly use the reward function $r$ in the version space refinement. Instead, we use a truncated reward function $r_{\mathcal{U}^k}$ that is zero for any $(h, s, a, b, s')$ in the infrequent transition set $\mathcal{U}^k$. This truncated design is critical to our analysis and the subsequent guarantees, e.g., see Lemma B.10. For the truncated reward functions, the backup step in Algorithm 6 should be understood as: $\bar{Q}_h^\pi(s, a, b) = \mathbb{E}_{s' \sim \hat{P}_h^k(\cdot|s,a,b)} \left[ r_h(s, a, b) 1\{(h, s, a, b, s') \notin \mathcal{U}^k\} + \bar{V}_{h+1}^\pi(s') \right], \forall (s, a, b)$.

We now present a theoretical guarantee for APE-OVE. We bound policy regret in terms of an instance-dependent quantity, namely minimum positive visitation probability, defined as follows.

---

[4]A $\sqrt{H}$ factor in $H\sqrt{AB}$ is perhaps unrelated to the need to learn the opponent's moves. This factor perhaps can be removed with a more intricate algorithm that takes into account the variance of transition kernels.

---

**Algorithm 3** Adaptive Policy Elimination by Optimistic Value Estimation (APE-OVE)

---

1: **Input**: number of episodes $T$, reward function $r$
2: **Parameters**: $\alpha := \log(\mathcal{N}_\Theta(1/T)HSAT/\delta), \bar{T} := \min\{t \in \mathbb{N} : (m-1)\log\log t + t \geq \frac{T}{HSAB}\}$,
   $K = \mathcal{O}(\log\log \bar{T})$, and $T_k := \bar{T}^{1-\frac{1}{2^k}}, \forall k \in [K]$
3: **Initialize**: $\Pi^1 = \Pi, \Theta^1 = \Theta$
4: **for** epoch $k = 1, \ldots, K$ **do**
5: $\quad$ $\hat{P}^k, \Theta^k, \mathcal{U}^k = \text{LAYERWISE\_EXPLORATION}(\Pi^k, T_k)$ (Algorithm 4)
6: $\quad$ $\Pi^{k+1} := \left\{\pi \in \Pi^k : \bar{V}^\pi(r_{\mathcal{U}^k}, \hat{P}^k, \Theta^k) \geq \max_{\pi \in \Pi^k} \bar{V}^\pi(r_{\mathcal{U}^k}, \hat{P}^k, \Theta^k) - cH^2SAB\sqrt{\alpha/(d^*T_k)}\right\}$
   $\quad$ where $\quad r_{\mathcal{U}^k}(s_1, a_1, b_1, \ldots, s_H, a_H, b_H) \quad := \quad \sum_{h\in[H]} 1\{(h, s_h, a_h, b_h, s_{h+1}) \quad \notin$
   $\quad \mathcal{U}^k\}r_h(s_h, a_h, b_h)$ and $\bar{V}^\pi(r, P, \Theta) := \text{OPTIMISTIC\_VALUE\_ESTIMATE}(\pi, r, P, \Theta)$ is
   $\quad$ given in Algorithm 6
7: **end for**

---

**Algorithm 4** LAYERWISE\_EXPLORATION$(\Pi^k, T_k)$

---

1: **Input**: Policy version space $\Pi^k$, number of episodes $T_k$
2: **Initialize**: $\hat{P}^k = \{\hat{P}_h^k\}_{h\in[H]}$ arbitrary transition kernels, $\mathcal{U}^k = \emptyset, \Theta_{hsa}^k = \Theta, \forall(h, s, a), \mathcal{D} = \emptyset$,
   $N_h^k(s, a, b, s') = 0, \forall(h, s, a, b, s')$, and for each $(h, s, a, b), 1_{hsab}$ is the reward function $r'$ such
   $r_{h'}'(s', a', b') = 1\{(h', s', a', b') = (h, s, a, b)\}$
3: **for** $h = 1, \ldots, H$ **do**
4: $\quad$ **for** $(s, a, b) \in \mathcal{S} \times \mathcal{A} \times \mathcal{B}$ **do**
5: $\quad\quad$ $\pi^{khsab} = \arg\max_{\pi\in\Pi^k} \text{OPTIMISTIC\_VALUE\_ESTIMATE}(\pi, 1_{hsab}, \hat{P}^k, \Theta^k)$ (Algorithm 6)
6: $\quad\quad$ Play $\pi^{khsab}$ for $m - 1$ episodes (and collect nothing)
7: $\quad\quad$ Keep playing $\pi^{khsab}$ for $T_k$ episodes and add all the transitions only at step $h$ to $\mathcal{D}$
8: $\quad$ **end for**
9: $\quad$ $N_h^k(s, a, b, s') \leftarrow N_h^k(s, a, b, s') + 1, \forall(s, a, b, s')$ s.t. $(h, s, a, b, s') \in \mathcal{D}$
10: $\quad$ $\Theta_{hsa}^k = \{\theta \in \Theta_{hsa}^k : \sum_{b:(h,s,a,b)\in\mathcal{D}} P_\theta(b) \geq \max_{\theta\in\Theta_{hsa}^k} \sum_{b:(h,s,a,b)\in\mathcal{D}} P_\theta(b) - \alpha\}, \forall(s, a) \in \mathcal{S} \times \mathcal{A}$
11: $\quad$ $\mathcal{U}^k \leftarrow \mathcal{U}^k \cup \{(h, s, a, b, s') : N_h^k(h, s, a, b, s') \leq cH^2\log(SABHK/\delta)\}$
12: $\quad$ $\hat{P}_h^k = \text{TRANSITION\_ESTIMATE}(h, N_h^k, \mathcal{U}^k, s^\dagger)$ (Algorithm 5)
13: $\quad$ Reset $\mathcal{D} = \emptyset$
14: **end for**
15: **Output**: $\hat{P}^k = \{\hat{P}_h^k\}_{h\in[H]}, \Theta^k, \mathcal{U}^k$

---

**Definition 4** (Minimum positive visitation probability). *The quantity $d^* := \inf_{h,s,a:d_h^*(s,a)>0} d_h^*(s, a)$ is said to be the minimum positive visitation probability, where $d_h^*(s, a) := \inf_{\pi\in\Pi:d_h^{\pi,f([\pi]^m)}(s,a)>0} d_h^{\pi,f([\pi]^m)}(s, a)$.*

The minimum positive visitation probability – which has also been used recently to characterize instance-dependent bounds for PAC RL [Tirinzoni et al., 2023], is the minimal probability that any state-action pair can be visited at a time step, given they can be visited at all. This implies that during the exploration phase if we try a certain policy $\pi$ for $N$ episodes and encounter $(s, a)$ at step $h$ (in any episode), on average, $\pi$ would visit $(h, s, a)$ for $Nd^*$ times out of $N$ episodes. This, along with the assumption that the adversary is consistent enables us to estimate the adversary's response to any $(h, s, a)$ that is visited within an estimation error of order $1/\sqrt{Nd^*}$. Note that we do not need to take care of the adversary's response to any $(h, s, a)$ that is not visited as these tuples are deemed infrequent by any policy and thus have negligible impact on the value estimation.

**Theorem 5.** *Playing APE-OVE against any $m$-memory bounded, stationary, and consistent adversaries in any Markov game for $T$ episodes, with $T = \tilde{\Omega}(\max\{\frac{H^5AB(d^*)^2}{S^3}, (m-1)HSAB\})$, and*

$$T \gtrsim \min\{\frac{H^5SAB(d^*)^2\log^4(HSABK/\delta)}{\alpha^2}, \frac{H^9(d^*)^2\log^4(HSABK/\delta)}{(SAB)^3\alpha^2}, \frac{H^{13}\log^2(HSABK/\delta)}{(AB)^3S^5}\},$$

---

**Algorithm 5** TRANSITION_ESTIMATE$(h, N_h, \mathcal{U}, s^\dagger)$

1: $\hat{P}_h(s'|s,a,b) = \begin{cases} \frac{N_h(s,a,b,s')}{N_h(s,a,b)}, \forall (s,a,b,s') \text{ s.t. } (h,s,a,b,s') \notin \mathcal{U} \\ 0, \forall (s,a,b,s') \text{ s.t. } (h,s,a,b,s') \in \mathcal{U} \end{cases}$

2: $\hat{P}_h(s^\dagger|s,a,b) = 1 - \sum_{s' \in \mathcal{S}:(h,s,a,b,s') \notin \mathcal{U}} \hat{P}_h(s'|s,a,b), \forall (s,a,b) \in \mathcal{S} \times \mathcal{A} \times \mathcal{B}$

3: $\hat{P}_h(s^\dagger|s^\dagger,a,b) = 1, \forall (a,b) \in \mathcal{A} \times \mathcal{B}$

4: **Output**: $\hat{P}_h$

---

**Algorithm 6** OPTIMISTIC_VALUE_ESTIMATE$(\pi, r, P, \Theta)$

1: **Input**: reward function $r$, policy $\pi$, transition kernel $P$, parameter version space $\Theta$
2: **Initialize**: $\bar{V}^\pi_{H+1}(\cdot) = 0$
3: **for** $h = H, H-1, \ldots, 1$ **do**
4: $\quad \bar{Q}^\pi_h(s,a,b) = r_h(s,a,b) + [P_h \bar{V}^\pi_{h+1}](s,a,b), \forall (s,a,b)$
5: $\quad \bar{V}^\pi_h(s) = \max_{\theta \in \Theta_{hs\pi_h(s)}} \bar{Q}^\pi_h(s, \pi_h(s), P_\theta), \forall s$ $\qquad\qquad\qquad$ ▷ *Optimistic MLE*
6: **end for**
7: **Output**: $\bar{V}^\pi_1(s_1)$

---

*guarantees that with probability at least $1 - \delta$,*

$$PR(T) = \mathcal{O}\left((m-1)H^2 SAB \log\log T + H^{3/2}\sqrt{SAB}(HSAB + H^2 + S^{3/2}AB)\sqrt{\frac{T\alpha}{d^*}}\log\log T\right)$$

*where $d^*$ is the minimum positive visitation probability and $\alpha$ is as defined in Algorithm 3.*

Theorem 5 asserts a $\sqrt{T}$ policy regret bound against $m$-memory bounded, stationary, and consistent adversaries in Markov games. Notably, our bounds grow linearly with memory length $m$. Compared to the bound in Theorem 4, given $T$ is sufficiently large, the bound in Theorem 5 deals with the general memory length $m$ at the cost of a worse dependence on all other factors $H, S, A, B, d^*$. Dealing with $\zeta$-approximately consistent adversaries (see Remark 1) will incur an additional term $\mathcal{O}(T\zeta)$ to the policy regret.

## 6 Discussion

In this paper, we study learning in Markov games against adaptive adversaries and highlight the statistical hardness of learning in this setting. We identify a natural structural assumption on the response function of the adversary, wherein we provide two distinct algorithms that attain $\sqrt{T}$ policy regret, one for the unit memory and the other for general memory length.

There are several notable gaps in our current understanding of policy regret in Markov games. First, we do not know if the dependence on the minimum positive visitation probability $d^*$ when learning against $m$-memory bounded opponents is necessary. In other words, can we derive minimax bounds that hold for any problem instance, regardless of how small $d^*$ is, for the case of general $m$? While it seems to us that such a dependence is necessary (as it seems difficult otherwise to learn the opponent's response while also learning high-return policies), yet we are unable to prove or reject this conjecture. Second, as we state in Remark 2, we do not currently know the necessary conditions on the opponent's response functions for learnability in this setting. This might as well require an alternate condition that generalizes our notion of consistent behaviors and fully characterizes the predictability of the opponent (in a similar way as the VC dimension characterizes learnability in statistical learning theory). Third, our theory currently views information, and not computation, as the main bottleneck and aims for policy regret minimization without worrying about computational complexity. As a result, some of the steps in our algorithms happen to be computationally inefficient. In particular, selecting a policy that maximizes the optimistic value function requires iterating over the learner's policy set, which is exponentially large. Can we hope for computationally efficient no-policy regret algorithms in Markov games? Fourth, our policy regret bounds scale with the cardinality of the state space and the action space, which could be large in many practical settings. Can we avoid such dependence by employing function approximation (e.g., neural networks)?

## Acknowledgments and Disclosure of Funding

This research was supported, in part, by the DARPA GARD award HR00112020004, NSF CAREER award IIS-1943251, funding from the Institute for Assured Autonomy (IAA) at JHU, and the Spring'22 workshop on "Learning and Games" at the Simons Institute for the Theory of Computing.

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

# Contents

# A   Missing proofs for Section 4

## A.1   Proof of Theorem 1

*Proof of Theorem 1.* The construction of a hard problem essentially follows the proof idea of Arora et al. [2012]. Policy regret requires the learner to compete with the best fixed sequence of policy in hindsight as if she could have changed her past policies. The lower bound utilizes this fact to construct an instance such that once the learner picks a particular policy in the first episode, she will receive a low reward for the remaining episodes. The only way to achieve a higher reward is to go back in time and select a different policy.

More formally, let's consider any learner. Let $\pi^1$ be a policy that the learner commits in the first episode with the highest positive probability $p > 0$. Note that $\pi^1$ and $p$ are the inherent property of the learner and *do not* depend on the adversary and the Markov game as in the first episode, the learner has zero information about the adversary and the Markov game. Now let's consider the adversary that depends only on the learner's policy in the first episode and nothing else, i.e., for all $t$ and policy sequence $\pi^1, \ldots, \pi^t$, $f_t(\pi^1, \ldots, \pi^t) = f(\pi^1)$ for some function $f : \Pi \rightarrow \Psi$. In addition, let $f$ such that $f(\pi) = \mu$ if $\pi = \pi^1$ and $f(\pi) = \nu$ otherwise, where $\mu$ and $\nu$ such that for all $s$, $\sup_{\pi \neq \pi^1} V_1^{\pi,\nu}(s) - \sup_\pi V_1^{\pi,\mu}(s) = \Omega(1)$. There exists a Markov game that always guarantees the existence of such $\mu, \nu$ (the constructions are fairly straightforward). Thus, with probability $p$, we have $\text{PR}(T) = \Omega(T)$. Note that the external regret $R(T)$ for this construction is 0.

$\square$

## A.2   Proof of Theorem 2

*Proof of Theorem 2.* The proof follows from the two main arguments: (i) a reduction from any latent MDP [Kwon et al., 2021] to a Markov game with an adversary playing policies from a finite set of Markov policies, and (ii) a reduction from the notion of regret in latent MDPs to the policy regret w.r.t. an oblivious sequence of Markov policies.

Argument (i) is directly taken from [Liu et al., 2022, Proposition 5]. In particular, interacting with any latent MDP [Kwon et al., 2021] of $L$ latent variables, $S$ states, $A$ actions, $H$ time steps, and binary rewards is equivalent to interacting (from the perspective of the learner) a (simulated) Markov game against an adversary whose policies are chosen from a set of $L$ Markov policies. In particular, the simulated Markov game has $SA + S$ states, $A$ actions for the learner, $2S$ actions for the adversary, and $2H$ time steps (see [Liu et al., 2022, Section A.4] for the detailed construction of the simulated Markov game from any latent MDP). Thus, we can utilize any lower bound for latent MDP for the Markov game (but not vice versa).

To continue from Argument (i) and begin with Argument (ii), we recall the definition of latent MDPs [Kwon et al., 2021]. At the beginning of each episode, the nature secretly draws uniformly at random from a set of $L$ base MDPs and the learner interacts with this drawn MDP for the episode. [Kwon et al., 2021, Theorem 3.1] show that for any learner, there exists a latent MDP with $L$ base MDPs such that the learner needs at least $\Omega((SA/L)^L/\epsilon^2)$ episodes to identify an $\epsilon$-suboptimal policy, where the optimality is defined with respect to the average values over the $M$ base MDPs. Note that in the construction of the hard latent MDP instance above, there is a unique optimal policy (let's call it $\pi^*$) with respect to the aforementioned optimality notion. Thus, the regret of this learner over $T$ episodes competing against $\pi^*$ is at least $\Omega(\sqrt{T(SA/L)^L})$ (the learner suffers an instantaneous regret of $\epsilon$ every time she fails to identify $\pi^*$). Note again that the regret above is the expectation with respect to the uniform distribution over $L$ base MDPs. Thus, there exists a particular realization of a sequence of $T$ base MDPs in a certain order such that the regret with respect to this sequence when competing with $\pi^*$ is at least the expected regret with respect to the uniform distribution over $L$ base MDPs, which is $\Omega(\sqrt{T(SA/L)^L})$. Finally, note that $\pi^*$ is also an optimal policy with respect to the total value across the sequence of $T$ MDPs since $\pi^*$ is an optimal policy for each individual base MDP, per the construction in Kwon et al. [2021]. Thus, we can conclude that, for any learner, there exists a sequence of $T$ MDPs from a set of $L$ MDPs such that the regret of the learner with respect to this MDP sequence is $\Omega(\sqrt{T(SA/L)^L})$.

$\square$

### A.3 Proof of Theorem 3

*Proof of Theorem 3.* Consider any learner. Consider the adversary's policy space $\Psi = \{\mu, \nu\}$ where for all $h \in [H-1]$, $\mu_h$ and $\nu_h$ are arbitrary but $\mu_H(b_1|s) = 1, \forall s$ and $\nu_H(b_2|s) = 1, \forall s$, for some $b_1, b_2 \in \mathcal{B}$. Let the reactive function $f$ to map all policies but some $\pi^*$ in $\Pi$ to $\mu$, whereas $f(\pi^*) = \nu$. Now consider a deterministic Markov game with the following properties. The transition kernel is deterministic and always traverses through the same sequence of states, regardless of what actions the learner and the adversary take. The reward functions are deterministic everywhere, and also zero everywhere except that $r_H(s, a, b_2) = 1, \forall s, a$. Except for $\pi^*$ that yields a positive reward if the learner selects it, all other policies in $\Pi$ give zero reward. In addition, since the learner does not know $f$ and that there is no relation whatsoever between $f(\pi)$ and $f(\pi')$ for any $\pi \neq \pi'$, the learner needs to play all policies in $\Pi$ at least once to be able to identify $\pi^*$. $\qquad\square$

## B Missing proofs for Section 5

### B.1 Support lemmas

**Maximum Likelihood Estimation.** Let $\{x_i\}_{i \in [T]} \sim P_{\theta^*}$ where $\theta^* \in \Theta$. Denote $\mathcal{N}_\Theta(\epsilon)$ the $\epsilon$-bracketing number of function class $\{P_\theta : \theta \in \Theta\}$. The following lemma says that the log-likelihood of the true model in the empirical data is close to that of any model within the model class, up to an error that scales logarithmically with the model complexity measured in a bracketing number.

**Lemma B.1.** *There exists an absolute constant $c$ such that for any $\delta \in (0, 1)$, with probability at least $1 - \delta$, for all $t \in [T]$ and $\theta \in \Theta$, we have*

$$\sum_{i=1}^{t} \log \frac{P_\theta(x_i)}{P_{\theta^*}(x_i)} \le c \log(\mathcal{N}_\Theta(1/T)T/\delta).$$

The following lemma says that any model that is close to the true model in the log-likelihood in the historical data would yield a similar data distribution as the true model.

**Lemma B.2.** *There exists an absolute constant $c$ such that for any $\delta \in (0, 1)$, with probability at least $1 - \delta$, for all $t \in [T]$ and $\theta \in \Theta$,*

$$d_{TV}^2(P_\theta, P_{\theta^*}) \le \frac{c}{t}\left(\sum_{i=1}^{t} \log \frac{P_{\theta^*}(x_i)}{P_\theta(x_i)} + \log(\mathcal{N}_\Theta(1/T)T/\delta)\right),$$

*where $d_{TV}$ denotes the total variation distance.*

The two lemmas above directly follow from [Liu et al., 2023, Proposition B.1] and [Liu et al., 2023, Proposition B.2], respectively, wherein the analysis built on the classical analysis of MLE [Geer, 2000] and the "tangent" sequence analysis in [Zhang, 2006, Agarwal et al., 2020], respectively. The following lemma is a direct corollary of Lemma B.1 and Lemma B.2.

**Lemma B.3.** *Let $\hat{\theta}_t \in \arg\sup_{\theta \in \Theta} \sum_{i=1}^{t} \log P_\theta(x_i)$. Define the version space:*

$$\Theta_t := \left\{\theta \in \Theta : \sum_{i=1}^{t} \log P_\theta(x_i) \ge \sum_{i=1}^{t} \log P_{\hat{\theta}_t}(x_i) - c \log(\mathcal{N}_\Theta(1/T)T/\delta)\right\}.$$

*Then, with probability at least $1 - \delta$, for all $t \in [T]$, we have $\theta^* \in \Theta_t$ and*

$$\max_{\theta \in \Theta_t} d_{TV}(P_\theta, P_{\theta^*}) \le c\sqrt{\frac{\log(\mathcal{N}_\Theta(1/T)T/\delta)}{t}}.$$

### B.2 Proof of Theorem 4

We first introduce several notations that we will use throughout our proofs. We denote $N_h^t$ and $\Theta_i^t$ the counters $N_h$ and the parameter confidence sets $\Theta_i^t$ at the beginning of the episode $t$.

**Lemma B.4** (Optimism). *With probability at least $1 - \delta$, for all $(h, s, \pi, t)$, we have*

$$\bar{V}_h^\pi(s) \geq V_h^{\pi, f(\pi)}(s).$$

*Proof of Lemma B.4.* We will prove a stronger statement: For any $(h, s, a, b, \pi)$, we have

$$\bar{Q}_h^\pi(s, a, b) \geq Q_h^{\pi, f(\pi)}(s, a, b) \text{ and } \bar{V}_h^\pi(s) \geq V_h^{\pi, f(\pi)}(s).$$

We will prove by induction with $h \in [H + 1]$. For $h = H + 1$, the claim in the lemma trivially holds. Assume by induction that the claim holds for some $h + 1$. We will prove that it holds for $h$. Indeed, for any $(s, a, b)$ such that $\bar{Q}_h^\pi(s, a, b) = H - h + 1$, of course $\bar{Q}_h^\pi(s, a, b) \geq Q_h^{\pi, f(\pi)}(s, a, b)$. Consider any $(s, a, b)$ such that $\bar{Q}_h^\pi(s, a, b) < H - h + 1$, we have

$$
\begin{aligned}
\bar{Q}_h^\pi(s, a, b) - Q_h^{\pi, f(\pi)}(s, a, b) &= [\hat{P}_h \bar{V}_{h+1}^\pi](s, a, b) + r_h(s, a, b) + \beta(N_h(s, a, b)) \\
&\quad - ([P_h V_{h+1}^{\pi, f(\pi)}](s, a, b) + r_h(s, a, b)) \\
&\geq [(\hat{P}_h - P_h)V_{h+1}^{\pi, f(\pi)}](s, a, b) + \beta(N_h(s, a, b)) \\
&\geq 0,
\end{aligned}
$$

where the first inequality uses the induction assumption that $\bar{V}_{h+1}^\pi \geq V_{h+1}^{\pi, f(\pi)}$ and the last inequality uses Hoeffding's inequality and the union bound. In addition, it follows from Lemma B.3 and the union bound that, with probability at least $1 - \delta$, for any $(t, h, s, \pi)$, we have

$$f(\pi)_h(\cdot|s) \in P_{\Theta_{hs\pi_h(s)}^t}$$

Under the same event wherein the above relation holds, we have

$$
\begin{aligned}
\bar{V}_h^\pi(s) &= \max_{\theta \in \Theta_{hs\pi_h(s)}} \bar{Q}_h^\pi(s, \pi_h(s), P_\theta) \\
&\geq \bar{Q}_h^\pi(s, \pi_h(s), f(\pi)_h) \\
&\geq Q_h^{\pi, f(\pi)}(s, \pi_h(s), f(\pi)_h) \\
&= V_h^{\pi, f(\pi)}(s).
\end{aligned}
$$

This completes the case for $h + 1$ and thus completes the proof. $\qquad\square$

*Proof of Theorem 4.* By the optimism of $\bar{V}$ (Lemma B.4), with probability at least $1 - \delta$, for all $(t, \pi)$, we have

$$V_1^{\pi, f(\pi)}(s_1^t) - V_1^{\pi^t, f(\pi^t)}(s_1^t) \leq \bar{V}_1^\pi(s_1^t) - V_1^{\pi^t, f(\pi^t)}(s_1^t) \leq \bar{V}_1^{\pi^t}(s_1^t) - V_1^{\pi^t, f(\pi^t)}(s_1^t) = \Delta_1^t$$

where the second inequality follows from Line 4 of Algorithm 1, and the last equation is a result of what we now define:

$$\Delta_h^t := \bar{V}_h^{\pi^t}(s_h^t) - V_h^{\pi^t, f(\pi^t)}(s_h^t), \forall(t, h).$$

We now decompose $\Delta_h^t$ as follows:

$$
\begin{aligned}
\Delta_h^t &= \max_{\theta \in \Theta_{hs_h^t a_h^t}^t} \bar{Q}_h^{\pi^t}(s_h^t, a_h^t, P_\theta) - Q_h^{\pi^t, f(\pi^t)}(s_h^t, a_h^t, f(\pi^t)_h) \\
&= \underbrace{\bar{Q}_h^{\pi^t}(s_h^t, a_h^t, b_h^t) - Q_h^{\pi^t, f(\pi^t)}(s_h^t, a_h^t, b_h^t)}_{=:\xi_h^t} \\
&\quad + \underbrace{\bar{Q}_h^{\pi^t}(s_h^t, a_h^t, f(\pi^t)_h) - \bar{Q}_h^{\pi^t}(s_h^t, a_h^t, b_h^t) + Q_h^{\pi^t, f(\pi^t)}(s_h^t, a_h^t, b_h^t) - Q_h^{\pi^t, f(\pi^t)}(s_h^t, a_h^t, f(\pi^t)_h)}_{=:\zeta_h^t} \\
&\quad + \underbrace{\max_{\theta \in \Theta_{hs_h^t a_h^t}^t} \bar{Q}_h^{\pi^t}(s_h^t, a_h^t, P_\theta) - \bar{Q}_h^{\pi^t}(s_h^t, a_h^t, f(\pi^t)_h)}_{=:\gamma_h^t}.
\end{aligned}
$$

We will bound each of $\xi_h^t, \zeta_h^t, \gamma_h^t$ separately as follows.

**Bounding $\{\xi_h^t\}$.** For simplicity, we denote $x_h^t = (s_h^t, a_h^t, b_h^t)$. We define

$$V_{h+1}^*(s) = \sup_{\pi \in \Pi} V_{h+1}^{\pi, f(\pi)}(s), \forall s.$$

Note that the optimality above does not require that there exists an optimal policy $\pi^*$ such that $V_h^*(s) = V_h^{\pi^*, f(\pi^*)}(s), \forall (h, s)$. Note that if $\bar{Q}_h^{\pi^t}(x_h^t) = H - h + 1$, it is trivial that $\zeta_h^t \leq 0$. Thus, we only need to consider when $\bar{Q}_h^{\pi^t}(x_h^t) < H - h + 1$, and thus

$$
\begin{aligned}
\zeta_h^t &= [\hat{P}_h^t \bar{V}_{h+1}^{\pi^t}](x_h^t) + \beta(N_h^t(x_h^t)) - [P_h V_{h+1}^{\pi^t, f(\pi^t)}](x_h^t) \\
&= [(\hat{P}_h^t - P_h)V_{h+1}^*](x_h^t) + [(\hat{P}_h^t - P_h)(\bar{V}_{h+1}^{\pi^t} - V_{h+1}^*)](x_h^t) + [P_h(\bar{V}_{h+1}^{\pi^t} - V_{h+1}^{\pi^t, f(\pi^t)})](x_h^t) + \beta(N_h^t(x_h^t)) \\
&\leq [(\hat{P}_h^t - P_h)(\bar{V}_{h+1}^{\pi^t} - V_{h+1}^*)](x_h^t) + [P_h(\bar{V}_{h+1}^{\pi^t} - V_{h+1}^{\pi^t, f(\pi^t)})](x_h^t) + 2\beta(N_h^t(x_h^t)).
\end{aligned}
$$

By Bernstein's inequality, with probability at least $1 - \delta$, for all $(s, a, b, s', h, t)$ and with $\iota := \log(2S^2 ABHT/\delta)$, we have

$$
\begin{aligned}
\hat{P}_h^t(s'|s, a, b) - P_h(s'|s, a, b) &\leq \frac{\iota}{N_h^t(s, a, b)} + \sqrt{\frac{2P_h(s'|s, a, b)\iota}{N_h^t(s, a, b)}} \\
&\leq \frac{1}{H}P_h(s'|s, a, b) + \frac{H\iota}{2N_h^t(s, a, b)} + \frac{\iota}{N_h^t(s, a, b)} \\
&= \frac{1}{H}P_h(s'|s, a, b) + (1 + \frac{H}{2})\frac{\iota}{N_h^t(s, a, b)},
\end{aligned}
$$

where note that the first inequality holds even when $N_h^t(s, a, b) = 0$ and the second inequality follows form AM-GM. Thus, with probability $1 - \delta$, for all $(t, h)$, we have

$$[(\hat{P}_h^t - P_h)(\bar{V}_{h+1}^{\pi^t} - V_{h+1}^*)](x_h^t) \leq \frac{SH(1 + H/2)\iota}{N_h^t(x_h^t)} + \frac{1}{H}[P_h(\bar{V}_{h+1}^{\pi^t} - V_{h+1}^*)](x_h^t).$$

Plugging this inequality into $\zeta_h^t$ above, then with probability at least $1 - \delta$, for all $(t, h)$,

$$
\begin{aligned}
\zeta_h^t &\leq \frac{SH(1 + H/2)\iota}{N_h^t(x_h^t)} + (1 + \frac{1}{H})\left((\bar{V}_{h+1}^{\pi^t} - V_{h+1}^{\pi^t, f(\pi^t)})(s_{h+1}^t) + \epsilon_{h+1}^t\right) + 2\beta(N_h^t(x_h^t)) \\
&\leq \frac{3SH^2\iota}{2N_h^t(x_h^t)} + (1 + \frac{1}{H})\left(\Delta_{h+1}^t + \epsilon_{h+1}^t\right) + 2\beta(N_h^t(x_h^t)),
\end{aligned}
$$

where we define

$$\epsilon_{h+1}^t := [P_h(\bar{V}_{h+1}^{\pi^t} - V_{h+1}^{\pi^t, f(\pi^t)})](x_h^t) - (\bar{V}_{h+1}^{\pi^t} - V_{h+1}^{\pi^t, f(\pi^t)})(s_{h+1}^t).$$

**Bounding $\sum_t \zeta_h^t$ and $\sum_t \epsilon_h^t$.** Note that for all $h$, $\{\zeta_h^t\}_{t \in [T]}$ and $\{\epsilon_h^t\}_{t \in [T]}$ are martingale difference sequences. Thus, by Azuma-Hoeffding's inequality and the union bound, with probability at least $1 - \delta$, we have

$$\sum_{t,h} \zeta_h^t \lesssim H^2\sqrt{T\log(H/\delta)}, \text{ and } \sum_{t,h} \epsilon_h^t \lesssim H^2\sqrt{T\log(H/\delta)}$$

**Bounding $\{\gamma_h^t\}$.** By [Lemma B.3](), and the union bound, with probability at least $1 - \delta$, for all $(t, h)$, we have

$$
\begin{aligned}
\gamma_h^t &= \max_{\theta \in \Theta_{h s_h^t a_h^t}^t} \bar{Q}_h^{\pi^t}(s_h^t, a_h^t, P_\theta) - \bar{Q}_h^{\pi^t}(s_h^t, a_h^t, f(\pi^t)_h) \\
&\leq 2H \max_{\theta \in \Theta_{h s_h^t a_h^t}^t} d_{TV}(P_\theta, f(\pi^t)_h(\cdot|s_h^t)) \\
&\lesssim H\sqrt{\frac{\alpha}{N_h^t(s_h^t, a_h^t)}}.
\end{aligned}
$$

Plugging these bounds into the definition of $\Delta_h^t$, combining them using the union bound and re-scaling $\delta$, we have that: with probability at least $1 - \delta$, for all $(t, h, \pi)$, we have

$$\Delta_h^t = \xi_h^t + \zeta_h^t + \gamma_h^t$$

$$\lesssim \frac{3SH^2\iota}{2N_h^t(x_h^t)} + (1 + \frac{1}{H})\left(\Delta_{h+1}^t + \epsilon_{h+1}^t\right) + 2\beta(N_h^t(x_h^t)) + \zeta_h^t + H\sqrt{\frac{\alpha}{N_h^t(s_h^t, a_h^t)}}.$$

Thus, we have with probability at least $1 - \delta$, we have

$$\sum_{t=1}^{T}\Delta_1^t \lesssim \sum_{t=1}^{T}(1 + \frac{1}{H})^H\sum_{h=1}^{H}\left(\frac{3SH^2\iota}{2N_h^t(x_h^t)} + \epsilon_{h+1}^t + \zeta_h^t + 2\beta(N_h^t(x_h^t)) + H\sqrt{\frac{\alpha}{N_h^t(s_h^t, a_h^t)}}\right)$$

$$\lesssim SH^2\iota\sum_{t,h}\frac{1}{N_h^t(x_h^t)} + H^2\sqrt{T\log(H/\delta)} + H\sqrt{\log(HSABT|\Pi|/\delta)}\sum_{t,h}\frac{1}{\sqrt{N_h^t(x_h^t)}}$$

$$+ H\sqrt{\alpha}\sum_{t,h}\frac{1}{\sqrt{N_h^t(s_h^t, a_h^t)}}.$$

Finally, note that

$$\sum_{t,h}\frac{1}{N_h^t(x_h^t)} = \sum_{h}\sum_{(s,a,b)}\sum_{i=1}^{N_h^T(s,a,b)}\frac{1}{i} \leq \sum_{h}\sum_{(s,a,b):N_h^T(s,a,b)\geq 1}\log N_h^T(s,a,b) \leq HSAB\log T.$$

$$\sum_{t,h}\frac{1}{\sqrt{N_h^t(x_h^t)}} = \sum_{h}\sum_{(s,a,b)}\sum_{i=1}^{N_h^T(s,a,b)}\frac{1}{\sqrt{i}} \leq \sum_{h}\sum_{(s,a,b)}\sqrt{N_h^T(s,a,b)} \leq \sqrt{HSAB}\sqrt{\sum_{(h,s,a,b)}N_h^T(s,a,b)}$$

$$= H\sqrt{SABT}.$$

$$\sum_{t,h}\frac{1}{\sqrt{N_h^t(s_h^t, a_h^t)}} = \sum_{(h,s,a)}\sum_{i=1}^{N_h^T(s,a)}\frac{1}{\sqrt{i}} \leq \sum_{h,s,a}\sqrt{N_h^T(s,a)} \leq \sqrt{HSA}\sqrt{\sum_{h,s,a}N_h^T(s,a)} = H\sqrt{SAT}.$$

Plugging these three inequalities above into the bound for $\sum_{t=1}^{T}\Delta_1^t$ right before and re-scaling $\delta$ complete the proof. $\square$

### B.3 Proof of Theorem 5

The layerwise exploration stage (Algorithm 4) performs layerwise exploration for each layer $h \in [H]$ and estimates infrequent transitions into $\mathcal{U}$. Since infrequent transitions do not significantly affect policy evaluation in any way (will be proved precisely later), we can exclude them and quickly refrain from exploring them extensively. However, excluding them changes the underlying data distribution of the experiences that the earner would receive when interacting with the environment. To handle this bias issue, it is often convenient to consider an "absorbing" Markov game $M'$, a refinement of the original Markov game $M$ that excludes all infrequent transitions.

**Definition 5** (Absorbing Markov games). *Given a Markov game $M = (\mathcal{S}, \mathcal{A}, \mathcal{B}, r, P, H)$, a set of transitions $\mathcal{U}$, and a dummy state $s^\dagger$, an absorbing Markov game $M' = (\mathcal{S} \cup \{s^\dagger\}, \mathcal{A}, \mathcal{B}, r, \tilde{P}, H)$ w.r.t. $(M, \mathcal{U}, s^\dagger)$ is defined as follows: For any $(h, s, a, b, s') \in [H] \times \mathcal{S} \times \mathcal{A} \times \mathcal{B} \times \mathcal{S}$,*

$$\tilde{P}_h(s'|s, a, b) = \begin{cases} P_h(s'|s, a, b) & \text{if } (h, s, a, b) \notin \mathcal{U} \\ 0 & \text{if } (h, s, a, b) \in \mathcal{U}, \end{cases}$$

*$\tilde{P}_h(s^\dagger|s, a, b) = 1 - \sum_{s'\in\mathcal{S}}\tilde{P}_h(s'|s, a, b)$ and $\tilde{P}_h(s^\dagger|s^\dagger, a, b) = 1$. In addition, $r_h(s, a, b) = \begin{cases} r_h(s, a, b) \text{ if } s \in \mathcal{S} \\ 0 \text{ if } s = s^\dagger, \end{cases}$, $\pi_h(\cdot|s) = \begin{cases} \pi_h(\cdot|s) \text{ if } s \in \mathcal{S}, \\ \text{arbitrary if } s = s^\dagger, \end{cases}$, $\mu_h(\cdot|s) = \begin{cases} \mu_h(\cdot|s) \text{ if } s \in \mathcal{S}, \\ \text{arbitrary if } s = s^\dagger. \end{cases}$*

Let $\tilde{P}^k$ be the absorbing transition kernels w.r.t. $(M, \mathcal{U}^k, s^\dagger)$ (Definition 5). Notice that the transition dynamics $\hat{P}^k$ by Algorithm 4 are unbiased estimates of the absorbing transition dynamics $\tilde{P}$.

### B.3.1 Sampling policies are sufficiently exploratory

We now show that the sampling policies in the reward-free exploration stage are sufficiently exploratory over the state-action space of the Markov game. We start with bounding the difference between $\tilde{P}$ and $\hat{P}^k$ (Line 5 of Algorithm 3) using empirical Bernstein's inequality.

**Lemma B.5.** *Define the event E:* $\forall (k, h, s, a, b, s') \in [K] \times [H] \times \mathcal{S} \times \mathcal{A} \times \mathcal{B} \times \mathcal{S}$ *such that* $(h, s, a, b, s') \notin \mathcal{U}$,

$$|\hat{P}_h^k(s'|s, a, b) - \tilde{P}_h^k(s'|s, a, b)| \leq \sqrt{\frac{2\hat{P}_h^k(s'|s, a, b)\iota}{N_h^k(s, a, b)}} + \frac{7\iota}{3N_h^k(s, a, b)}$$

*where* $\iota := c\log(SABHK/\delta)$ *and* $N_h^k$ *are the counter at layer* $h$ *in epoch* $k$ *obtained at Line 9 by running Algorithm 4 in epoch* $k$. *Then, we have* $\Pr(E) \geq 1 - \delta$. *In addition,* $\forall (h, s, a, b, s') \in \mathcal{U}$, $\hat{P}_h^k(s'|s, a, b) = \tilde{P}_h^k(s'|s, a, b) = 0$.

*Proof of Lemma B.5.* Lemma B.5 is essentially the analogous of [Qiao et al., 2022, Lemma E.2] from MDPs to Markov games. The first part follows from empirical Bernstein's inequality and union bound. The second part comes from the definition of the absorbing transition kernels $\tilde{P}$ and the construction of the empirical transition kernels $\hat{P}^k$. $\qquad\square$

**Lemma B.6.** *Conditioned on the event E in Lemma B.5: For all* $(k, h, s, a, b, s') \in [K] \times [H] \times \mathcal{S} \times \mathcal{A} \times \mathcal{B} \times \mathcal{S}$ *such that* $(h, s, a, b, s') \notin \mathcal{U}$, *we have*

$$(1 - \frac{1}{H})\hat{P}_h^k(s'|s, a, b) \leq \tilde{P}_h^k(s'|s, a, b) \leq (1 + \frac{1}{H})\hat{P}_h^k(s'|s, a, b).$$

*Proof of Lemma B.6.* Lemma B.6 is essentially the same as [Qiao et al., 2022, Lemma E.3]. $\qquad\square$

**Lemma B.7.** *Conditioned on the event E in Lemma B.5: For all* $(k, h, s, a, b, s') \in [K] \times [H] \times \mathcal{S} \times \mathcal{A} \times \mathcal{B} \times \mathcal{S}$ *and any policy* $\pi$, *we have*

$$\frac{1}{4}V^{\pi, f([\pi]^m)}(1_{hsab}, \hat{P}^k) \leq V^{\pi, f([\pi]^m)}(1_{hsab}, \tilde{P}^k) \leq 3V^{\pi, f([\pi]^m)}(1_{hsab}, \hat{P}^k),$$

*where* $V^{\pi, \mu}(r, P)$ *denotes the expected total reward under policies* $(\pi, \mu)$ *and the Markov game specified by the reward function* $r$ *and transition kernels* $P$.

*Proof of Lemma B.7.* The proof essentially follows from the proof of [Qiao et al., 2022, Lemma E.5]. $\qquad\square$

Lemma B.5 to Lemma B.7 are similar in nature with corresponding lemmas in a single-agent MDP in [Qiao et al., 2022]. We now prove a novel lemma that's absent in the single-agent MDP setting yet crucial to our theorem. Recall our notion that, $\bar{V}^\pi(r, P, \Theta) :=$ OPTIMISTIC_VALUE_ESTIMATE$(\pi, r, P, \Theta)$ which is given in Algorithm 6.

**Lemma B.8.** *Fix any* $k \in [K]$ *and consider* $\hat{P}^k, \Theta^k, \mathcal{U}^k = LAYERWISE\_EXPLORATION(\Pi^k, T_k)$ *(Line 5 of Algorithm 3). Define the event* $E_k$: *for all* $(h, s, a, b) \in [H] \times \mathcal{S} \times \mathcal{A} \times \mathcal{B}$ *and all* $\pi \in \Pi$, *we have*

$$0 \leq \bar{V}^\pi(1_{hsab}, \hat{P}^k, \Theta^k) - V^{\pi, f([\pi]^m)}(1_{hsab}, \hat{P}^k) \leq \xi_{MLE}(T_k),$$

*where* $\xi_{MLE}(T_k) := cH\sqrt{\frac{\alpha}{d^* T_k}}$. *Assume that* $T$ *is sufficiently large such that* $T_k \geq \frac{2\log(SHKA/\delta)}{d^{*2}}, \forall k \in [K]$. *Then,* $\Pr(E_k) \geq 1 - \delta$.

*Proof of Lemma B.8.* Let us fix any $(h, s, a, b)$ and $\pi$. Note that the value function for any policy under any dynamic w.r.t. the reward function $1_{hsab}$ is zero at any step $h' > h$. Also, notice that, prior to the exploration of layer $h$ in the reward-free exploration (Algorithm 4), $\hat{P}_1^k, \ldots, \hat{P}_{h-1}^k$ are already constructed.

**Additional notations.** In OPTIMISTIC_VALUE_ESTIMATE($1_{hsab}, \hat{P}^k, \pi, \Theta$) (Algorithm 6), we denote the intermediate value estimates $\bar{V}_l^\pi$ by $\bar{V}_l^\pi(\cdot; 1_{hsab}, \hat{P}^k, \Theta^k)$ to emphasize the dependence on the reward function and the transition dynamics being used. We denote $N_h^k(s,a)$ the count of pairs $(h,s,a)$ during the $h$-th layer exploration of Algorithm 4. We write $V_h^{\pi,\mu}(s; r, P)$ in place of $V_h^{\pi,\mu}(s)$ to emphasize the dependence on the reward function $r$ and transition dynamic $P$.

We will evaluate the quantity $\Delta_l(\bar{s}) := V_l^{\pi, f([\pi]^m)}(\bar{s}; 1_{hsab}, \hat{P}^k) - \bar{V}_l^\pi(\bar{s}; 1_{hsab}, \hat{P}^k, \Theta^k)$ for any $l \in [h-1], \bar{s} \in \mathcal{S}$.

The first part $\bar{V}^\pi(1_{hsab}, \hat{P}^k, \Theta^k) - V^{\pi, f([\pi]^m)}(1_{hsab}, \hat{P}^k) \geq 0$ follows from that with probability at least $1 - \delta$, $\theta_{hsa}^* \in \Theta_{hsa}^k, \forall(h,s,a)$. Thus, $\bar{V}^\pi(1_{hsab}, \hat{P}^k, \Theta^k)$ is an optimistic estimate of $V^{\pi, f([\pi]^m)}(1_{hsab}, \hat{P}^k)$. For the second part, we have

$$
\begin{aligned}
\Delta_l(\bar{s}) &= \sup_{\theta \in \Theta_{l,\bar{s},\pi_l(\bar{s})}^k} \sum_{s' \in \mathcal{S}} P_l^k(s'|\bar{s}, \pi_l(\bar{s}), P_\theta) \bar{V}_{l+1}^\pi(s'; 1_{hsab}, \hat{P}^k, \Theta^k) \\
&\quad - \sum_{s' \in \mathcal{S}} P_l^k(s'|\bar{s}, \pi_l(\bar{s}), f([\pi]^m)_l(\cdot|\bar{s})) V_{l+1}^{\pi, f([\pi]^m)}(s'; 1_{hasb}, \hat{P}^k) \\
&= \sum_{s' \in \mathcal{S}} P_l^k(s'|\bar{s}, \pi_l(\bar{s}), f([\pi]^m)_l(\cdot|\bar{s})) \Delta_{l+1}(s') \\
&\quad + \sum_{s' \in \mathcal{S}} \left( P_l^k(s'|\bar{s}, \pi_l(\bar{s}), P_\theta) - P_l^k(s'|\bar{s}, \pi_l(\bar{s}), f([\pi]^m)_l(\cdot|\bar{s})) \right) \bar{V}_{l+1}^\pi(s'; 1_{hsab}, \hat{P}^k, \Theta^k) \\
&\leq \max\{\Delta_{l+1}(s') : s' \in \mathcal{S} \text{ s.t. } \exists b' \in \mathcal{B}, (l, \bar{s}, \pi_l(\bar{s}), b', s') \notin \mathcal{U}^k\} \\
&\quad + 1\{N_l^k(\bar{s}, \pi_l(\bar{s})) \geq 1\} \cdot 2 \max_{\theta \in \Theta_{l\bar{s}\pi_l(\bar{s})}^k} d_{TV}(f([\pi]^m)_l(\cdot|\bar{s}), P_\theta),
\end{aligned}
$$

where we use the convention that $\max \emptyset = 0$, and the last inequality follows from that $P_l^k(s'|\bar{s}, \pi_l(\bar{s}), b') = 0$ if $(l, \bar{s}, \pi_l(\bar{s}), b', s') \notin \mathcal{U}^k$, that $\bar{V}_{l+1}^\pi(s'; 1_{hsab}, \hat{P}^k, \Theta^k) \in [0,1]$, and that, for any two distributions $p, q \in [0,1]^{|\mathcal{X}|}$ over a finite support $\mathcal{X}$, we have $d_{TV}(p,q) = \frac{1}{2}\|p - q\|_1$.

If $N_l^k(\bar{s}, \pi_l(\bar{s})) = 0$, then $\Delta_l(\bar{s}) = 0$. Consider the case $N_l^k(\bar{s}, \pi_l(\bar{s})) \geq 1$. That means that *the state-action pair $(\bar{s}, \pi_l(\bar{s}))$ must be visited in step $l$ at least once by at least one policy $\pi^{kl\tilde{s}\tilde{a}\tilde{b}}$ for some $(\tilde{s}, \tilde{a}, \tilde{b}) \in \mathcal{S} \times \mathcal{A} \times \mathcal{B}$.* Note that this policy $\pi^{kl\tilde{s}\tilde{a}\tilde{b}}$ is run for $m - 1 + T_k$ consecutive episodes. Thus, by the definition of the minimum positive visitation probability $d^*$, we must have

$$
\mathbb{E}\left[N_l^k(\bar{s}, \pi_l(\bar{s}))\right] \geq d^* T_k,
$$

where the expectation is w.r.t. the transition kernel $P$ of the original Markov game $M$ and policy $\pi^{kl\tilde{s}\tilde{a}\tilde{b}}$. By Hoelfding's inequality and the union bound: With probability at least $1 - \delta$, for all $l, \bar{s}, k, \pi$, we have

$$
N_l^k(\bar{s}, \pi_l(\bar{s})) \geq \mathbb{E}\left[N_l^k(\bar{s}, \pi_l(\bar{s}))\right] - \sqrt{T_k \log(SHKA/\delta)}.
$$

In particular, for $(l, \bar{s}, k, \pi)$ such that $\mathbb{E}\left[N_l^k(\bar{s}, \pi_l(\bar{s}))\right] \geq d^* T_k$ and for $T_k \geq \frac{2\log(SHKA/\delta)}{d^{*2}}$, we have $N_l^k(\bar{s}, \pi_l(\bar{s})) \geq \frac{1}{2} d^* T_k$ with probability at least $1 - \delta$. Combined with Lemma B.3, with probability at least $1 - \delta$, we have

$$
\max_{\theta \in \Theta_{l\bar{s}\pi_l(\bar{s})}^k} d_{TV}(f([\pi]^m)_l(\cdot|\bar{s}), P_\theta) \leq c\sqrt{\frac{\alpha}{d^* T_k}}. \tag{2}
$$

Thus, under the same event that the above inequality holds, we have

$$
\Delta_1(s_1) \leq cH\sqrt{\frac{\alpha}{d^* T_k}}.
$$

$\square$

Next, we will show that the transition samples collected in $\mathcal{U}^k$ are indeed infrequent transitions by any policy. Let $\tau = (s_1, a_1, b_1, \ldots, s_H, a_H, b_H)$ be a random trajectory generated by the learner's policy $\pi$ and the opponent's policies $f([\pi]^m)$ for some policy $\pi$.

**Definition 6** (Bad events). *Under the original transition kernel $P$, we define $\mathcal{F}$ to be the event that there exists $h \in [H]$ such that $(h, s_h, a_h, b_h, s_{h+1}) \in \mathcal{U}^k$ and we define $\mathcal{F}_h$ to be the event such that $h$ is the smallest step that $(h, s_h, a_h, b_h, s_{h+1}) \in \mathcal{U}^k$. Under the absorbing transition kernel $\tilde{P}^k$, we define $\mathcal{F}$ to be the event that there exists $h \in [H]$ such that $s_{h+1} = s^\dagger$ and we define $\mathcal{F}_h$ to be the event such that $h$ is the smallest step that $s_{h+1} = s^\dagger$.*

**Lemma B.9.** *Conditioned on the event $E$ in Lemma B.5 and the event $E_k$ in Lemma B.8, with probability at least $1 - \delta$, for any $k \in [K]$, we have*

$$
\sup_{\pi \in \Pi^k} \Pr[\mathcal{F}|P, \pi] \lesssim \frac{H^3 \log(HSABK/\delta)}{T_k} + H\xi_{MLE}(T_k).
$$

*where $\xi_{MLE}(\cdot)$ is defined in Lemma B.8 and $\mathcal{F}$ is defined in Definition 6.*

*Proof of Lemma B.9.* Under the event $E$ in Lemma B.5 and the event $E_k$ in Lemma B.8, for any $(h, s, a, b)$, we have

$$
\begin{aligned}
V^{\pi^{khsab}, f([\pi^{khsab}]^m)}(1_{hsab}, \tilde{P}^k) &\geq \frac{1}{4} V^{\pi^{khsab}, f([\pi^{khsab}]^m)}(1_{hsab}, \hat{P}^k) \\
&\geq \frac{1}{4} \bar{V}^{\pi^{khsab}}(1_{hsab}, \hat{P}^k, \Theta^k) - \xi_{MLE}(T_k) \\
&= \frac{1}{4} \sup_{\pi \in \Pi^k} \bar{V}^{\pi}(1_{hsab}, \hat{P}^k, \Theta^k) - \xi_{MLE}(T_k) \\
&\geq \frac{1}{4} \sup_{\pi \in \Pi^k} V^{\pi, f([\pi]^m)}(1_{hsab}, \hat{P}^k) - \xi_{MLE}(T_k) \\
&\geq \frac{1}{12} \sup_{\pi \in \Pi^k} V^{\pi, f([\pi]^m)}(1_{hsab}, \tilde{P}^k) - \xi_{MLE}(T_k) \qquad (3)
\end{aligned}
$$

where the first inequality and the last inequality follow from Lemma B.7, the second inequality follows from Lemma B.8, the second and third inequality follow from Lemma B.8, and the equation follows from the definition of $\pi^{khsab}$ in Algorithm 4. Let $\pi^{kh}$ be a policy that chooses each $\pi^{khsab}$ with probability $\frac{1}{SAB}$ for any $(s, a, b) \in \mathcal{S} \times \mathcal{A} \times \mathcal{B}$. Thus, we have

$$
\begin{aligned}
\Pr[\mathcal{F}_h|P, \pi^{kh}] &= \Pr[\mathcal{F}_h|\tilde{P}^k, \pi^{kh}] \\
&= \frac{1}{SAB} \sum_{\bar{s}, \bar{a}, \bar{b}} \sum_{s,a,b} V^{\pi^{kh\bar{s}\bar{a}\bar{b}}, f([\pi^{kh\bar{s}\bar{a}\bar{b}}]^m)}(1_{hsab}, \tilde{P}^k) \tilde{P}_h(s^\dagger|s, a, b) \\
&\geq \frac{1}{SAB} \sum_{s,a,b} V^{\pi^{khsab}, f([\pi^{khsab}]^m)}(1_{hsab}, \tilde{P}^k) \tilde{P}_h(s^\dagger|s, a, b) \\
&\geq \frac{1}{12SAB} \sum_{s,a,b} \sup_{\pi \in \Pi^k} V^{\pi, f([\pi]^m)}(1_{hsab}, \tilde{P}^k) - \frac{1}{SAB}\xi_{MLE}(T_k) \\
&\geq \frac{1}{12SAB} \sup_{\pi \in \Pi^k} \sum_{s,a,b} V^{\pi, f([\pi]^m)}(1_{hsab}, \tilde{P}^k) - \frac{1}{SAB}\xi_{MLE}(T_k) \\
&= \frac{1}{12SAB} \sup_{\pi \in \Pi^k} \Pr[\mathcal{F}_h|\tilde{P}^k, \pi] - \frac{1}{SAB}\xi_{MLE}(T_k) \\
&= \frac{1}{12SAB} \sup_{\pi \in \Pi^k} \Pr[\mathcal{F}_h|P, \pi] - \frac{1}{SAB}\xi_{MLE}(T_k). \qquad (4)
\end{aligned}
$$

By the construction of $\mathcal{U}^k$, we have that

$$
\Pr[\mathcal{F}_h|P, \pi^{kh}] \leq c\frac{H^2 \log(HSABK/\delta)}{SABT_k}.
$$

Thus, combined with Equation (4), we have

$$\sup_{\pi \in \Pi^k} \Pr[\mathcal{F}_h | P, \pi] \lesssim \frac{H^2 \log(HSABK/\delta)}{T_k} + \xi_{MLE}(T_k).$$

Finally, note that

$$\Pr[\mathcal{F}|P, \pi] = \sum_{h \in [H]} \Pr[\mathcal{F}_h | P, \pi],$$

which concludes our proof.

$\square$

### B.3.2 Uniform policy evaluation

In this part, we will show that the empirical transition kernel $P^k$ constructed from the exploratory data by our sampling policies is a good surrogate for the true transition kernel $P$ in evaluating the value of uniformly all policies.

**Lemma B.10.** *Conditioned on the event $E$ in Lemma B.5 and the event $E_k$ in Lemma B.8 and the high-probability event in Lemma B.9, with probability at least $1 - \delta$, for any $k \in [K]$, any reward function $r$, and any policy $\pi \in \Pi^k$, we have*

$$0 \le V^{\pi, f([\pi]^m)}(r, P) - V^{\pi, f([\pi]^m)}(r_{\mathcal{U}^k}, \tilde{P}^k) \lesssim \frac{H^4 \log(HSABK/\delta)}{T_k} + H^2 \xi_{MLE}(T_k),$$

*where for any trajectory $\tau = (s_1, a_1, b_1, \ldots, s_H, a_H, b_H)$, $r_{\mathcal{U}^k}(\tau) := \sum_{h=1}^{H} 1\{(h, s_h, a_h, b_h, s_{h+1}) \notin \mathcal{U}^k)\} r_h(s_h, a_h, b_h)$.*

*Proof of Lemma B.10.* We have

$$
\begin{aligned}
V^{\pi, f([\pi]^m)}(r, P) &= \sum_{\tau} r(\tau) \Pr(\tau | P, \pi) \\
&= \sum_{\tau \notin \mathcal{F}} r(\tau) \Pr(\tau | P, \pi) + \sum_{\tau \in \mathcal{F}} r(\tau) \Pr(\tau | P, \pi) \\
&= \sum_{\tau \notin \mathcal{F}} r(\tau) \Pr(\tau | \tilde{P}^k, \pi) + \sum_{\tau \in \mathcal{F}} r(\tau) \Pr(\tau | P, \pi) \\
&= \sum_{\tau \notin \mathcal{F}} r_{\mathcal{U}^k}(\tau) \Pr(\tau | \tilde{P}^k, \pi) + \sum_{\tau \in \mathcal{F}} r(\tau) \Pr(\tau | P, \pi) \\
&\le V^{\pi, f([\pi]^m)}(r_{\mathcal{U}^k}, \tilde{P}^k) + \sum_{\tau \in \mathcal{F}} r(\tau) \Pr(\tau | P, \pi) \\
&\lesssim V^{\pi, f([\pi]^m)}(r_{\mathcal{U}^k}, \tilde{P}^k) + \frac{H^4 \log(HSABK/\delta)}{T_k} + H^2 \xi_{MLE}(T_k),
\end{aligned}
$$

where the last inequality is due to Lemma B.9. Similarly, we have

$$
\begin{aligned}
V^{\pi, f([\pi]^m)}(r, P) &= \sum_{\tau \notin \mathcal{F}} r_{\mathcal{U}^k}(\tau) \Pr(\tau | \tilde{P}^k, \pi) + \sum_{\tau \in \mathcal{F}} r(\tau) \Pr(\tau | P, \pi) \\
&\ge \sum_{\tau \notin \mathcal{F}} r_{\mathcal{U}^k}(\tau) \Pr(\tau | \tilde{P}^k, \pi) + \sum_{\tau \in \mathcal{F}} r_{\mathcal{U}^k}(\tau) \Pr(\tau | P, \pi) \\
&\ge \sum_{\tau \notin \mathcal{F}} r_{\mathcal{U}^k}(\tau) \Pr(\tau | \tilde{P}^k, \pi) + \sum_{\tau \in \mathcal{F}} r_{\mathcal{U}^k}(\tau) \Pr(\tau | \tilde{P}^k, \pi) \\
&= V^{\pi, f([\pi]^m)}(r_{\mathcal{U}^k}, \tilde{P}^k).
\end{aligned}
$$

$\square$

**Lemma B.11.** *With probability at least $1 - \delta$, for any $k \in [K]$, any reward function $r$, and any policy $\pi \in \Pi$, we have*

$$0 \leq \bar{V}^\pi(r_{\mathcal{U}^k}, \hat{P}^k, \Theta^k) - V^{\pi, f([\pi]^m)}(r_{\mathcal{U}^k}, \hat{P}^k) \lesssim H^2 \sqrt{\frac{\alpha}{d^* T_k}}.$$

*Proof of Lemma B.11.* The first inequality is trivial, following the first part of Lemma B.8. We will focus on the second inequality. Fix any deterministic policy $\pi$. For simplicity, we write $\bar{V}_h^\pi(s) := \bar{V}_h^\pi(r_{\mathcal{U}^k}, \hat{P}^k, \Theta^k)(s)$, and $V_h^\pi(s) := V_h^{\pi, f([\pi]^m)}(r_{\mathcal{U}^k}, \hat{P}^k)(s)$. Let $\Delta_h^k(s) := \bar{V}_h^\pi(s) - V_h^\pi(s)$.

First of all, by construction of $\hat{P}^k$ and $r_{\mathcal{U}^k}$, we have $\Delta_h^k(s) = 0$ if $s = s^\dagger$ or if $N_h^k(s, \pi_h(s)) = 0$. This explains the very reason we design the truncated reward function $r_{\mathcal{U}^k}$.

We now consider $s \in \mathcal{S}$ such that $N_h^k(s, \pi_h(s)) > 0$. This condition, along with the consistent behavior and the minimum visitation probability, allows us to estimate the response $f([\pi]^m)_h(\cdot|s)$ sufficiently. In particular, $f([\pi]^m)_h(\cdot|s)$ depends only on the data obtained by visiting $(h, s, \pi_h(s))$ which is indeed visited at least $d^* T_k$ times, thus can be estimated up to an order of $1/\sqrt{d^* T_k}$ error. We have

$$
\begin{aligned}
\Delta_h^k(s) &= r_{\mathcal{U}^k, h}(s, \pi_h(s), P_\theta) + \hat{P}_h^k \bar{V}_{h+1}^\pi(s, \pi_h(s), P_\theta) \\
&\quad - r_{\mathcal{U}^k, h}(s, \pi_h(s), f([\pi]^m)_h(\cdot|s)) - \hat{P}_h^k V_{h+1}^\pi(s, \pi_h(s), f([\pi]^m)_h(\cdot|s)) \\
&= r_{\mathcal{U}^k, h}(s, \pi_h(s), P_\theta) - r_{\mathcal{U}^k, h}(s, \pi_h(s), f([\pi]^m)_h(\cdot|s)) \\
&\quad + \hat{P}_h^k (\bar{V}_{h+1}^\pi - V_{h+1}^\pi)(s, \pi_h(s), f([\pi]^m)_h(\cdot|s)) \\
&\quad + \hat{P}_h^k \bar{V}_{h+1}^\pi(s, \pi_h(s), P_\theta) - \hat{P}_h^k \bar{V}_{h+1}^\pi(s, \pi_h(s), f([\pi]^m)_h(\cdot|s)) \\
&\leq \sup_{\theta \in \Theta_{hs\pi_h(s)}^k} d_{TV}(P_\theta, f([\pi]^m)_h(\cdot|s)) \\
&\quad + \max\{\Delta_{h+1}^k(s') : s' \in \mathcal{S} \text{ s.t. } \exists b \in \mathcal{B}, (h, s, \pi_h(s), b, s') \notin \mathcal{U}^k\} \\
&\quad + H \sup_{\theta \in \Theta_{hs\pi_h(s)}^k} d_{TV}(P_\theta, f([\pi]^m)_h(\cdot|s)) \\
&= (H+1) \sup_{\theta \in \Theta_{hs\pi_h(s)}^k} d_{TV}(P_\theta, f([\pi]^m)_h(\cdot|s)) \\
&\quad + \max\{\Delta_{h+1}^k(s') : s' \in \mathcal{S} \text{ s.t. } \exists b \in \mathcal{B}, (h, s, \pi_h(s), b, s') \notin \mathcal{U}^k\}
\end{aligned}
$$

Note that, similar to Equation (2), as $N_h^k(s, \pi_h(s)) > 0$, with probability at least $1 - \delta$, we have

$$\sup_{\theta \in \Theta_{hs\pi_h(s)}^k} d_{TV}(P_\theta, f([\pi]^m)_h(\cdot|s)) \lesssim \sqrt{\frac{\alpha}{d^* T_k}}.$$

Thus, we have

$$\Delta_1^k(s_1) \lesssim H^2 \sqrt{\frac{\alpha}{d^* T_k}}.$$

$\square$

**Lemma B.12.** *Conditioned on the event $E$ in Lemma B.5 and the event $E_k$ in Lemma B.8, with probability $1 - \delta$, for any $k \in [K]$, $\pi \in \Pi$ and any reward function $r'$, we have*

$$|V^{\pi, f([\pi]^m)}(r', \hat{P}^k) - V^{\pi, f([\pi]^m)}(r', \tilde{P}^k)| \lesssim H S^{3/2} AB \sqrt{\frac{\log(HAT/\delta)}{T_k}} + HSAB \cdot \xi_{MLE}(T_k).$$

*Proof of Lemma B.12.* By the simulation lemma [Dann et al., 2017], we have

$$|V^{\pi, f([\pi]^m)}(r', \hat{P}^k) - V^{\pi, f([\pi]^m)}(r', \tilde{P}^k)| \leq \mathbb{E}_{\tilde{P}^k, \pi} \sum_{h=1}^H |(\hat{P}_h^k - \tilde{P}_h^k) \hat{V}_{h+1}^\pi|,$$

where $\hat{V}_{h+1}^\pi := V^{\pi,f([\pi]^m)}(r', \hat{P}^k)$. Define the sampling distribution $\nu_h \in \Delta(\mathcal{S} \times \mathcal{A} \times \mathcal{B}), h \in [H]$ by

$$\nu_h(s,a,b) := \frac{1}{SAB} \sum_{\bar{s},\bar{a},\bar{b}} V^{\pi^{kh\bar{s}\bar{a}\bar{b}},f([\pi^{kh\bar{s}\bar{a}\bar{b}}]^m)}(1_{hsab}, \tilde{P}^k). \tag{5}$$

Then, we have

$$\mathbb{E}_{\tilde{P}^k,\pi}|(\hat{P}_h^k - \tilde{P}_h^k)\hat{V}_{h+1}^\pi| = \sum_{s,a,b}|(\hat{P}_h^k - \tilde{P}_h^k)\hat{V}_{h+1}^\pi(s,a,b)| \cdot V^{\pi,f([\pi]^m)}(1_{hsab}, \tilde{P}^k)$$

$$= \sum_{s,a,b}|(\hat{P}_h^k - \tilde{P}_h^k)\hat{V}_{h+1}^\pi(s,a,b)| \cdot V^{\pi,f([\pi]^m)}(1_{hsab}, \tilde{P}^k)1\{\pi_h(s) = a\}$$

$$\leq 12\sum_{s,a,b}|(\hat{P}_h^k - \tilde{P}_h^k)\hat{V}_{h+1}^\pi(s,a,b)|1\{\pi_h(s) = a\} \cdot V^{\pi^{khsab},f([\pi^{khsab}]^m)}(1_{hsab}, \tilde{P}^k)$$

$$+ 12HSAB \cdot \xi_{MLE}(T_k)$$

$$\leq 12\sum_{s,a,b}|(\hat{P}_h^k - \tilde{P}_h^k)\hat{V}_{h+1}^\pi(s,a,b)|1\{\pi_h(s) = a\} \cdot \sum_{\bar{s},\bar{a},\bar{b}} V^{\pi^{kh\bar{s}\bar{a}\bar{b}},f([\pi^{kh\bar{s}\bar{a}\bar{b}}]^m)}(1_{hsab}, \tilde{P}^k)$$

$$+ 12HSAB \cdot \xi_{MLE}(T_k)$$

$$\leq 12SAB\sum_{s,a,b}|(\hat{P}_h^k - \tilde{P}_h^k)\hat{V}_{h+1}^\pi(s,a,b)|1\{\pi_h(s) = a\}\nu_h(s,a,b) + 12HSAB \cdot \xi_{MLE}(T_k)$$

$$= 12SAB\sqrt{\sum_{s,a,b}|(\hat{P}_h^k - \tilde{P}_h^k)\hat{V}_{h+1}^\pi(s,a,b)|^2\nu_h(s,a,b)1\{a = \pi_h(s)\}} + 12HSAB \cdot \xi_{MLE}(T_k)$$

$$\leq 12SAB\sqrt{\sup_{V:\mathcal{S}\cup s^\dagger \to [0,H]} \sup_{g:\mathcal{S}\cup s^\dagger \to \mathcal{A}} \mathbb{E}_{\nu_h}|(\hat{P}_h^k - \tilde{P}_h^k)V(s,a,b)|^21\{g(s) = a\}}$$

$$+ 12HSAB \cdot \xi_{MLE}(T_k)$$

$$\lesssim HS^{3/2}AB\sqrt{\frac{\log(HAT/\delta)}{T_k}} + HSAB \cdot \xi_{MLE}(T_k),$$

where the second equality is due to that $\pi$ is deterministic, the first inequality follows from [Equation (3)](#), the third inequality follows from Jensen's inequality, and the last inequality follows from the fundamental [Lemma B.13](#). $\qquad\square$

**Lemma B.13** ([Jin et al., 2020, Lemma C.2]). *With probability at least $1-\delta$, for all $h \in [H], k \in [K]$, we have*

$$\sup_{V:\mathcal{S}\cup s^\dagger \to [0,H]} \sup_{g:\mathcal{S}\cup s^\dagger \to \mathcal{A}} \mathbb{E}_{\nu_h}|(\hat{P}_h^k - \tilde{P}_h)V(s,a,b)|^21\{g(s) = a\} \lesssim \frac{H^2S\log(HAT/\delta)}{T_k}.$$

*Proof of [Lemma B.13](#).* Note that $\hat{P}^k$ is the empirical transition kernel constructed by sampling according to the data distribution $\nu$ defined in [Equation (5)](#) for $T_k$ samples, under the transition kernel $\tilde{P}^k$. Thus, [Lemma B.13](#) is a direct application of [Jin et al., 2020, Lemma C.2]. $\qquad\square$

**Lemma B.14.** *Let $\pi^* = \arg\max_{\pi \in \Pi} V^{\pi,f([\pi]^m)}$. Conditioned on the event of [Lemma B.10](#), [Lemma B.11](#), [Lemma B.12](#), $\pi^*$ never get eliminated from $\Pi^k$ for $k \in [K]$ by [Algorithm 3](#).*

*Proof of [Lemma B.14](#).* We will prove by induction. Since $\Pi^1$ contains all the possible policies of the learner, $\pi^* \in \Pi^1$. Assume by induction that $\pi^* \in \Pi^k$. We will show that $\pi^* \in \Pi^{k+1}$. Indeed, let $\hat{\pi}^k = \arg\max_{\pi \in \Pi^k} \bar{V}^\pi(r_{\mathcal{U}^k}, \hat{P}^k, \Theta^k)$, we have

$$\bar{V}^{\pi^*}(r_{\mathcal{U}^k}, \hat{P}^k, \Theta^k) \overset{\text{Lemma B.11}}{\geq} V^{\pi^*,f([\pi^*]^m)}(r_{\mathcal{U}^k}, \hat{P}^k)$$

$$\overset{\text{Lemma B.12}}{\gtrsim} V^{\pi^*,f([\pi^*]^m)}(r_{\mathcal{U}^k}, \tilde{P}^k) - HS^{3/2}AB\sqrt{\frac{\log(HAT/\delta)}{T_k}} - HSAB \cdot \xi_{MLE}(T_k)$$

$$\overset{\text{Lemma B.10}, \pi^* \in \Pi^k}{\geq} V^{\pi^*, f([\pi^*]^m)}(r, P) - HS^{3/2}AB\sqrt{\frac{\log(HAT/\delta)}{T_k}} - HSAB \cdot \xi_{MLE}(T_k)$$

$$- \frac{H^4 \log(HSABK/\delta)}{T_k} - H^2 \xi_{MLE}(T_k)$$

$$\overset{\pi^* \text{ is optimal}}{\geq} V^{\hat{\pi}^k, f([\hat{\pi}^k]^m)}(r, P) - HS^{3/2}AB\sqrt{\frac{\log(HAT/\delta)}{T_k}} - HSAB \cdot \xi_{MLE}(T_k)$$

$$- \frac{H^4 \log(HSABK/\delta)}{T_k} - H^2 \xi_{MLE}(T_k)$$

$$\overset{\text{Lemma B.10}, \pi^* \in \Pi^k}{\geq} V^{\hat{\pi}^k, f([\hat{\pi}^k]^m)}(r_{\mathcal{U}^k}, \tilde{P}^k) - HS^{3/2}AB\sqrt{\frac{\log(HAT/\delta)}{T_k}} - HSAB \cdot \xi_{MLE}(T_k)$$

$$- \frac{H^4 \log(HSABK/\delta)}{T_k} - H^2 \xi_{MLE}(T_k)$$

$$\overset{\text{Lemma B.12}}{\gtrsim} V^{\hat{\pi}^k, f([\hat{\pi}^k]^m)}(r_{\mathcal{U}^k}, \hat{P}^k) - HS^{3/2}AB\sqrt{\frac{\log(HAT/\delta)}{T_k}} - HSAB \cdot \xi_{MLE}(T_k)$$

$$- \frac{H^4 \log(HSABK/\delta)}{T_k} - H^2 \xi_{MLE}(T_k)$$

$$\overset{\text{Lemma B.11}}{\gtrsim} V^{\hat{\pi}^k, f([\hat{\pi}^k]^m)}(r_{\mathcal{U}^k}, \hat{P}^k, \Theta^k) - HS^{3/2}AB\sqrt{\frac{\log(HAT/\delta)}{T_k}} - HSAB \cdot \xi_{MLE}(T_k)$$

$$- \frac{H^4 \log(HSABK/\delta)}{T_k} - H^2 \xi_{MLE}(T_k) - H^2\sqrt{\frac{\alpha}{d^* T_k}}$$

$$\geq \max_{\pi \in \Pi^k} V^{\pi, f([\pi]^m)}(r_{\mathcal{U}^k}, \hat{P}^k, \Theta^k) - \mathcal{O}\left( H^2(SAB + H)\sqrt{\frac{\alpha}{d^* T_k}} + \frac{H^4 \log(HSABK/\delta)}{T_k} \right.$$

$$\left. + HS^{3/2}AB\sqrt{\frac{\log(HAT/\delta)}{T_k}} \right).$$

Thus, $\pi^* \in \Pi^{k+1}$. $\qquad\qquad \square$

Finally, we will show that any policy in $\Pi^{k+1}$ is of high quality.

**Lemma B.15.** *Recall the version space $\Pi^{k+1}$ defined at Line 6 of Algorithm 3. With probability at least $1 - \delta$, for any $k \in [K]$ and any $\pi \in \Pi^{k+1}$, and any reward function $r$, we have*

$$\sup_{\pi \in \Pi} V^{\pi, f([\pi]^m)}(r, P) - V^{\pi, f([\pi]^m)}(r, P) = \mathcal{O}\left( H^2(SAB + H)\sqrt{\frac{\alpha}{d^* T_k}} + \frac{H^4 \log(HSABK/\delta)}{T_k} \right.$$

$$\left. + HS^{3/2}AB\sqrt{\frac{\log(HAT/\delta)}{T_k}} \right).$$

*Proof of Lemma B.15.* Consider any $\pi \in \Pi^{k+1}$. We have

$$V^{\pi, f([\pi]^m)}(r, P) \geq V^{\pi, f([\pi]^m)}(r_{\mathcal{U}^k}, \tilde{P}^k)$$

$$\geq V^{\pi, f([\pi]^m)}(r_{\mathcal{U}^k}, \hat{P}^k) - HS^{3/2}AB\sqrt{\frac{\log(HAT/\delta)}{T_k}} - H^2 SAB\sqrt{\frac{\alpha}{d^* T_k}}$$

$$\geq \bar{V}^{\pi}(r_{\mathcal{U}^k}, \hat{P}^k, \Theta^k) - H^2\sqrt{\frac{\alpha}{d^* T_k}} - HS^{3/2}AB\sqrt{\frac{\log(HAT/\delta)}{T_k}} - H^2 SAB\sqrt{\frac{\alpha}{d^* T_k}}$$

$$\geq \sup_{\pi \in \Pi^k} \bar{V}^{\pi}(r_{\mathcal{U}^k}, \hat{P}^k, \Theta^k) - \mathcal{O}\left( H^2 SAB\sqrt{\frac{\alpha}{d^* T_k}} + HS^{3/2}AB\sqrt{\frac{\log(HAT/\delta)}{T_k}} \right)$$

$$\geq \sup_{\pi \in \Pi^k} V^{\pi, f([\pi]^m)}(r_{\mathcal{U}^k}, \hat{P}^k) - \mathcal{O}\left(H^2 SAB\sqrt{\frac{\alpha}{d^* T_k}} + HS^{3/2}AB\sqrt{\frac{\log(HAT/\delta)}{T_k}}\right)$$

$$\geq \sup_{\pi \in \Pi^k} V^{\pi, f([\pi]^m)}(r_{\mathcal{U}^k}, \tilde{P}^k) - \mathcal{O}\left(H^2 SAB\sqrt{\frac{\alpha}{d^* T_k}} + HS^{3/2}AB\sqrt{\frac{\log(HAT/\delta)}{T_k}}\right)$$

$$\geq \sup_{\pi \in \Pi^k} V^{\pi, f([\pi]^m)}(r, P)$$

$$- \mathcal{O}\left(H^2 SAB\sqrt{\frac{\alpha}{d^* T_k}} + HS^{3/2}AB\sqrt{\frac{\log(HAT/\delta)}{T_k}} + \frac{H^4 \log(HSABK/\delta)}{T_k} + H^3\sqrt{\frac{\alpha}{d^* T_k}}\right)$$

$$\geq \sup_{\pi \in \Pi} V^{\pi, f([\pi]^m)}(r, P)$$

$$- \mathcal{O}\left(H^2 SAB\sqrt{\frac{\alpha}{d^* T_k}} + HS^{3/2}AB\sqrt{\frac{\log(HAT/\delta)}{T_k}} + \frac{H^4 \log(HSABK/\delta)}{T_k} + H^3\sqrt{\frac{\alpha}{d^* T_k}}\right)$$

where the first inequality follows from the first part of Lemma B.10, the second inequality follows from Lemma B.12, the third inequality follows from Lemma B.11, the fourth inequality follows from the definition of $\Pi^{k+1}$, the fifth inequality follows from Lemma B.11, the sixth inequality follows from Lemma B.12, and the seventh inequality follows from the second part of Lemma B.10, and the last inequality follows from Lemma B.14. $\qquad\square$

*Proof of Theorem 5.* Note that $K = \min\{j : \sum_{k=1}^{j} T_k \geq \bar{T}\} = \mathcal{O}(\log\log\bar{T})$. Moreover, Algorithm 3 runs for $\sum_{k=1}^{K} HSAB(m - 1 + T_k) = T$ episodes, by the choice of $T_k = \bar{T}^{1-\frac{1}{2^k}}$, where $\bar{T} := \min\{t \in \mathbb{N} : (m-1)\log\log t + t \geq \frac{T}{HSAB}\}$. By Lemma B.15, with probability at least $1 - \delta$, we have

$$PR(T) \lesssim (m - 1 + T_1)H^2 SAB$$

$$+ \sum_{k=2}^{K}\left((m-1)H^2 SAB + HSAB \cdot T_k\left(H^2(SAB + H)\sqrt{\frac{\alpha}{d^* T_{k-1}}} + \frac{H^4 \log(HSABK/\delta)}{T_{k-1}}\right.\right.$$

$$\left.\left. + HS^{3/2}AB\sqrt{\frac{\log(HAT/\delta)}{T_{k-1}}}\right)\right)$$

$$\lesssim (m-1)H^2 SABK + H^2 SAB\sqrt{\bar{T}} + KH^3 SAB(SAB + H)\sqrt{\frac{\alpha\bar{T}}{d^*}}$$

$$+ H^5 SAB\log(HSABK/\delta)\sum_{k=2}^{K}\bar{T}^{\frac{1}{2^k}} + KH^2 S^{5/2}A^2 B^2\sqrt{\bar{T}\log(HAT/\delta)}$$

$$\lesssim (m-1)H^2 SABK + H^{3/2}\sqrt{SABT} + KH^{5/2}\sqrt{SAB}(SAB + H)\sqrt{\frac{\alpha T}{d^*}}$$

$$+ KH^{19/4}(SAB)^{3/4}\log(HSABK/\delta)T^{1/4} + K(HAB)^{3/2}S^2\sqrt{T\log(HAT/\delta)}$$

(because $\bar{T} \leq \frac{T}{HSAB}$).

Note that the third term always dominates the second term. We can further simplify the bound (in the last inequality above), by making either the third term or the last term dominate the fourth term, which is implied by,

$$T \gtrsim \min\{\frac{H^5 SAB(d^*)^2 \log^4(HSABK/\delta)}{\alpha^2}, \frac{H^9(d^*)^2 \log^4(HSABK/\delta)}{(SAB)^3 \alpha^2}, \frac{H^{13}\log^2(HSABK/\delta)}{(AB)^3 S^5}.\}$$

Also notice the condition $T_k \geq \frac{2\log(SHKA/\delta)}{d^{*2}}, \forall k \in [K]$ in Lemma B.8 translates into:

$$T \gtrsim \frac{HSAB\log^2(SHKA/\delta)}{(d^*)^4}.$$

Under these conditions of $T$, the bound becomes:

$$(m-1)H^2SABK + KH^{3/2}\sqrt{SAB}(HSAB + H^2 + S^{3/2}AB)\sqrt{\frac{T\alpha}{d^*}}.$$

$\square$

