# OpenReview forum: "Learning in Markov Games with Adaptive Adversaries: Policy Regret, Fundamental Barriers, and Efficient Algorithms"
_NeurIPS.cc/2024/Conference — NeurIPS 2024 poster_

### Official Review · Reviewer_qNT8 · 2024-06-24

**Soundness:** 3
**Presentation:** 2
**Contribution:** 3
**Rating:** 5
**Confidence:** 2

**Summary:**

This paper provides upper and lower bound on the complexity of learning Markov games. The authors focus on the notion of "policy regret", which already exist in bandit and repeated games and adapt this notion to the general case of episodic Markov games.

The first results of the authors are negative results: The authors shows that, in the general case, there are no algorithm achieveing a small regret (Theorem 1, 2 and 3 of the paper, for different variants of the problem).

Given these negative results, in a second part, the authors introduce a notion of "consistent adversaries" (Definition 3) and dervive an algorithm that has a small regret against this weak adversary.

**Strengths:**

Markov games are notoriously hard to learn. This paper presents some results to show how hard they are to learn in some specific settings.

The paper contains both negative and positive results on classes of problems that can be learned.

**Weaknesses:**

The paper is too dense, at the point that it hurts readability:
- the proofs of all results are only in the appendix and no intuition is give in the paper
- the algorithms are very hard to read because their presentation is too compact
- some parts of the algorithms are only given in appendix
As a result, the paper is essentially impossible to understand without looking at the appendix.

The notion of "consistent adversary" deserves more discussion: given that a policy is supposed to exploit future, I do not understand why it is consistent for the response of the adversary should essentially only depend on the action in a given state (which is sort of what this assumption implies).

The notion of policy regret is very strong, which illustrated by the fact that it is essentially impossible to have an algorithm with a low regret (Theorem 1, 2, 3) unless imposing very strong assumption on the learner (Theorem 4). Hence, this notion of regret feels a bit arbitrary to me.

The impossibility results are very similar to the one of [37].

**Questions:**

Please justify the regret notion and the notion of "constistent adversary".

**Limitations:**

NA.

---

> ### Author Rebuttal · Authors · 2024-08-05
>
> We thank the reviewer for the valuable feedback.
>
> ---
>
> > "The paper is too dense, ...  without looking at the appendix."
>
> While two subroutines for Algorithm 3 were put in the appendix, these are fairly standard subroutines and a mere distraction. In fact, we moved it for the sake of better readability – regardless, if one is unfamiliar with them, they can easily find it in the appendix (we give simple embedded hyperlinks for ease of navigation).
>
> We do discuss at length high-level intuitions behind every key result, idea and algorithm in the main text. The intuition and the details for the technical proofs are also presented in the appendix, which are not required for understanding our main results. It is worth noting that other reviewers commended us on the clarity of writing and presentation.
>
> ---
>
> > The notion of "consistent adversary"
>
> To be clear, consistency is a sufficient condition for learnability. In fact, we can weaken it to be approximately consistent (see discussion with Reviewer 1MKG). As we say on Lines 255-261, consistency implies that, given any $(h,s,a)$, the response $f([\pi]^m)_h(\cdot|s)$ stays the same across all policies $\pi$ such that $\pi_h(s) = a$. Thus, given any $(h,s)$, there are $A$ possible ways the adversary can play in step h at state s.
>
> Here is what we already say in the paper, but can emphasize it all in one place, per your suggestion.
>
> - (a) Adversary being consistent does not imply that it is sub-optimal. On the contrary, we assume that the adversary is all knowledgeable and has infinite computational power to have figured out the optimal response to any strategy of the learner. So being consistent does not take away any power from the adversary.
> - (b) A consistent adversary does not always play the same strategy when the learner adopts the same strategy at (s,h). The adversary’s strategy does not depend only on the learner's strategy at (s,h), but also what the learner played in previous (m-1) episodes.
> - (c) We agree that consistency may be too strong to seem necessary for ensuring learnability. We believe that something weaker, e.g., approximate consistency (i.e., small deviations) should still be fine. That could be an avenue for future work. What we have currently is still a very interesting result as it gives an easy to understand notion which is sufficient for learnability.
>
> ---
>
> > "The notion of policy regret is very strong ... Hence, this notion of regret feels a bit arbitrary to me."
>
>
> Regarding “Policy Regret it Arbitrary”, We could not disagree more with the reviewer. We argue that if the environment is adaptive/reactive, i.e., the actions of the opponent depend on learner’s past actions then no other notion of regret even makes sense. This is what has already been argued and settled in numerous papers before us. This is what motivated policy regret in the first place (see the paper by Arora et al. ICML 2012.) So, not the notion of policy regret is not arbitrary. If you want to do counterfactual learning, that is the only notion of regret that even makes sense. Of course, if you are in a setting where the environment is oblivious to your actions (e.g., forecasting weather), then you do not need policy regret – but most real world applications are not like that.
>
> For instance, consider the following scenario: a big investing firm unveils its new trading software, which implements the strategy of switching all of the firm’s investments back and forth between Google and Microsoft on a daily basis. This strategy sets off huge fluctuations in the stock market (the reactive environment), and the market crashes. A post hoc regret analysis reveals that any competing strategy (for instance, buying and holding Apple shares) would have lost all the money, too (as the market crashed), thereby making the regret zero and perhaps concluding that there was nothing wrong with the strategy. It is completely ignored that the market crashed in reaction to the algorithm’s actions and would have reacted differently to a different sequence of actions. Clearly, the notion of regret is misleading in a reactive scenario. In other words, minimizing regret against adaptive adversaries may not lead to learning. Policy regret, on the other hand, is a counterfactual notion of regret, which evaluates a competing strategy on the sequence of events that would have been generated if the competing strategy were followed. This is why policy regret is preferred (and only thing that makes sense) when you are up against a strategic opponent.
>
> Regarding “Hardness of minimizing policy regret”. Yes, minimizing policy regret is harder than standard regret since the adversary is given more power. This is why we ask what restrictions can we impose on the adversary so that it is stronger than oblivious adversaries but not completely arbitrary against whom it is impossible to learn. That is also the point of the hardness results. So, Theorems 1-3 are actually very useful results from a theory perspective. Negative results are as valuable as positive results (if not more), as they tell us what is not even possible, so we can stop wasting our time thinking about the problem in that way.
>
> Hopefully, the reviewer sees our contributions differently in the light of our comments above and will reconsider their evaluation of our work. Thanks!
>
> ---
>
>
>
> > The impossibility results are also very similar to the one of [37].
>
> Except in proof of Thm2 that benefits from a reduction of any latent MDP to a Markov game constructed by [37], our negative results are of completely different nature, and our paper and [37] also solve two completely different problems. Negative result in [37] is about external regret. Our negative results say that policy regret is linear in T or exponentially large when the adversary has unbounded memory or is non-stationary. Can you be more specific about how exactly similar our impossibility results are to the one of [37]?

---

> > ### Comment · Reviewer_qNT8 · 2024-08-12
> >
> > Thank you for your answer. I was probably too picky on oyvfirdt evaluation given the other papers that I had to review. I updated my score.

---

> > > ### Author Response · Authors · 2024-08-12
> > >
> > > We thank you and appreciate your response to our rebuttal. We find the rating a bit harsh given that we have addressed your concerns — we explained why policy regret is important and why negative results are important in theoretical works. We wonder if you can provide a clear reasoning/justification for your current rating. The other three reviewers all seem positive about the paper. If you have further questions, we are happy to address them.

---

### Official Review · Reviewer_1MKG · 2024-07-08

**Soundness:** 4
**Presentation:** 3
**Contribution:** 3
**Rating:** 7
**Confidence:** 3

**Summary:**

This paper studies learning in a dynamically evolving environment modeled as a Markov game (MG) and the adversarial is allowed to be adaptive. Authors focus on the policy regret rather than external regret commonly used by many existing work, and further investigate the fundamental limits on learning MG with different memory size. Finally, authors restrict adversary’s behavior and propose efficient algorithms achieving sublinear policy regret bounds.

**Strengths:**

- Studying how the power (w.r.t. memory and other behaviors characterized by stationarity and consistency) of adversary impacts the learnability is very interesting.

- Show statistically hardness for unbounded memory adversary and further show that even if the adversary is stationary, the hardness cannot be alleviated.

- Consistency of adversary is introduced and algorithms with sublinear policy regret are proposed for special case m=1 and general case m.

- The connections between existing results and related ones are clearly presented.

**Weaknesses:**

No major weaknesses. Other minor points have been well discussed in the paper.

**Questions:**

Apart from imposing additional constraints on the adversary, I am curious whether the proposed algorithm can smoothly degrade with the increase of inconsistency. Specifically, in definition 3, given two sequence of policies $\pi$ and $v$ with the same policy mapping, if we do not assume $f_t(\pi)$ and $f_t(v)$ are exactly the same, but assume the difference is denoted as $D_t$ and let $D=\sum_{t=1}^T D_t$ be the inconsistency. I’d appreciate if authors can discuss this.

**Limitations:**

Yes

---

> ### Author Rebuttal · Authors · 2024-08-05
>
> We thank the reviewer for the valuable feedback.
>
> ---
>
> > Question about "approximate consistency" $D$
>
> We thank the reviewer for an interesting suggestion. Our approach estimates the version space of the response function of the adversary using the adversary’s actions. The MLE analysis (Lemma B.1 and B.2) readily enables us to incorporate an inconsistency measure of D into the radius of the version space. Thus, our final policy regret bound should scale linearly with D as a result. We will discuss this extension in more detail in our revision. But we are happy to provide a more detailed analysis here, if the reviewer is interested.

---

> > ### Comment · Reviewer_1MKG · 2024-08-12
> > **Official Comment by Reviewer 1MKG**
> >
> > Thanks for your response. I will keep my score.

---

> > > ### Author Response · Authors · 2024-08-12
> > >
> > > Thanks for acknowledging our response!

---

### Official Review · Reviewer_yS7W · 2024-07-11

**Soundness:** 4
**Presentation:** 4
**Contribution:** 2
**Rating:** 6
**Confidence:** 3

**Summary:**

This paper addresses the problem of designing optimal strategies in Markov Games against adaptive adversaries. Specifically, the paper proposes the notion of $\textit{policy regret}$ which admits the adversary's ability to adaptively change their policies according to the policies applied by the learner. This work demonstrates statistically hardness results for the learner to achieve no-policy-regret when the adversary has unbounded memory or is non-stationary w.r.t different episodes. Further more, under the $\textit{consistent adversary}$ assumption, this work designs $O(\sqrt{T})$-regret algorithms respectively to the scenario when the memory bound for the adversary is 1 or some constant $m$. The algorithms proposed are some optimistic variants of upper confidence bound value iteration.

**Strengths:**

1. This paper move a step further from the previous results in [1] by introducing novel analysis w.r.t. policy regret.

2. Hardness results are shown when assumptions are not met, indicating the necessity of such assumptions.

3. Two proposed algorithms achieve $O(\sqrt{T})$ regret.

4. In general the paper is well-organized and theorems are well-supported by the proof.

[1] Qinghua Liu, Yuanhao Wang, and Chi Jin. Learning markov games with adversarial opponents: Efficient algorithms and fundamental limits. In International Conference on Machine Learning, pages 14036–14053. PMLR, 2022

**Weaknesses:**

My concerns and questions mainly focuses on the consistent adversary assumption.

1. While I understand the necessity of proposing some assumptions about the adversary in order for the learner to efficiently learn the adversarial strategy mapping function $f(\pi)$, this assumption doesn't not seem to be a favorable one. Firstly, looking from a game theoretical perspective, it makes sense even for a self-interested adversary to utilize previous information and play differently even when the learner adopts the same strategy at $(s, h)$. For example in a two-player zero-sum game, if the adversary observes that the learner behaves poorly at some certain state $s'$, then it is reasonable for the adversary to play certain strategy which leads to $(s', h+1)$ from $(s, h)$.

2. Furthermore, when the policy space for the learner includes all the deterministic strategies and when the adversary is consistent. The cardinality for the strategy space for the adversary is also finite, this is in sharp contrast with the results in [1] where only one of the two needs to be finite.

3. When the adversary is consistent, is it the case that the transition probability solely depends on the state and the learner? In other words, is this setting equivalent to the setting of single-controller Markov games where the controller is always the learner?

4. When the memory bound for the adversary is greater than 1, this paper proposed an algorithm which is a variant of [2] in order to achieve optimal global switching cost. However, under the consistent adversary assumption, would it be better to consider minimizing local switching cost instead?

[1] Qinghua Liu, Yuanhao Wang, and Chi Jin. Learning markov games with adversarial opponents: Efficient algorithms and fundamental limits. In International Conference on Machine Learning, pages 14036–14053. PMLR, 2022

[2] Dan Qiao, Ming Yin, Ming Min, and Yu-Xiang Wang. Sample-efficient reinforcement learning with loglog (t) switching cost. In International Conference on Machine Learning, pages 18031–18061. PMLR, 2022

**Questions:**

See weakness.

---

> ### Author Rebuttal · Authors · 2024-08-05
>
> We thank the reviewer for the valuable feedback.
>
> ---
>
> > "Weakness 1."
>
> There are three points we would like to make.
>
> - (a) The assumption that the adversary’s behavior is consistent does not imply that it is sub-optimal. On the contrary, we assume that the adversary is all knowledgeable and has infinite computational power to have figured out the optimal response to any strategy of the learner. So being consistent does not take away any power from the adversary.
>
> - (b) A consistent adversary does not always play the same strategy when the learner adopts the same strategy at (s,h). The adversary has memory; that is, the adversary’s strategy does not only depend on the learner’s strategy at (s,h), but also what the learner has adopted in the previous (m-1) episodes.
>
> - (c) We agree with the reviewer that consistent behavior may be too strong to seem necessary for ensuring learnability. In fact, we firmly believe that something weaker, for e.g., approximate consistency (i.e., small deviations) should still be fine. That seems far from trivial and could be an avenue for future work. What we have currently is still a very interesting result as it gives an easy to understand notion which is sufficient for learnability.
>
> ---
>
> > "Weakness 2."
>
> Actually, the policy space for the learner can be infinite with appropriate assumptions on its bounded complexity and the response function, but dealing with that largely deviates from the main points we want to convey in this paper. We assumed the learner’s policy space of all deterministic strategies for the sake of simplicity of presentation. More importantly, our paper and [1] are solving two completely different problems (policy regret minimization vs external regret minimization), making the two results incomparable. To be clear, we discussed [1] in our paper only to contrast the differences of the two settings, not to compare the two results.
>
> ---
>
> > "Weakness 3."
>
> No. The transition probability still depends on the adversary’s actions. Even when the adversary is consistent,  the adversary’s (*possibly non-memoryless*) response function f can still be arbitrary in any state. The learner does not know and cannot control the response function.
>
> ---
>
> > "Weakness 4."
>
> Thank you for the interesting question. However, we argue a low local switching cost algorithm would not be suitable for this case. The consistent behavior assumption has more to do with enabling an efficient estimation of the adversary’s response function, and less to do with the local switching cost behavior. That said, sample efficient algorithms could also be possible for a different structural assumption other than the consistent behavior assumption, as long as it is not necessary to visit all possible policies that the learner can take in order to learn about the adversary’s response behavior (as an extreme example, when $f$ maps all of the learner’s strategies to the same policy, which reduces the problem to single-agent MDP).
>
> We can, more formally, argue that low local switching cost cannot obtain low policy regret in general. Recall that the goal is to achieve low (policy) regret against the benchmark $ \max_{\pi} V_1^{\pi, f([\pi]^m)}$. A low local switching cost algorithm would only guarantee that a sequence of consecutive policies $\pi^1, …., \pi^{m-1}$ can be very similar “locally” (e.g., they agree in *many but all* states and steps) but they are not guaranteed to be identical “globally”. So, even in the episode $m$ that the learner happens to play the optimal policy $\pi^* = arg \max_{\pi} V_1^{\pi, f([\pi]^m)}$, the learner only gets to see the data for playing $\pi^*$ and $f(\pi^1, …, \pi^{m-1}, \pi^*)$. This data can be completely non informative for the purpose of estimating $V_1^{\pi^*, f([\pi^*]^m)}$ in the worst case, since $f(\pi^1, …, \pi^{m-1}, \pi^*)$ can be arbitrarily different from $f([\pi^*]^m)$, even when each $\pi^1, …, \pi^{m-1}$ might be similar to $\pi^*$ locally. For example, let’s say $m-1 \leq H$ and $\pi^i$ agrees with $\pi^*$ in all steps $h \neq i$. In this case, the consistent adversary places no restriction whatsoever in his response $f(\pi^1, …, \pi^{m-1}, \pi^*)$, i.e., $f(\pi^1, …, \pi^{m-1}, \pi^*)$ can literally be anything and disagree with $f([\pi^*]^m)$ in every single state and step.

---

> > ### Comment · Reviewer_yS7W · 2024-08-11
> >
> > Thanks for making clarifications, I've adjusted my score accordingly.

---

> > > ### Author Response · Authors · 2024-08-12
> > >
> > > Thanks for acknowledging our response!

---

### Official Review · Reviewer_Gp14 · 2024-07-15

**Soundness:** 3
**Presentation:** 3
**Contribution:** 3
**Rating:** 6
**Confidence:** 2

**Summary:**

The paper studies the learning problem in a Markov game against the adaptive adversary. The adversary's policy can depend on all the learner's past strategies. The paper first shows that if the adversary can be fully adaptive, then sublinear policy regret cannot be obtained for the learner. The paper then characterizes the fundamental barriers for the learner to achieve the sublinear regret. The paper shows that if the adversary is $m$-memory bounded, i.e., the adversary's strategy depends at most on the $m$ past strategies of the learner, then the sublinear policy regret is achievable. The paper provides both the lower bound and the efficient algorithm that achieves a regret upper bound at the tight $O(\sqrt{T})$ order.

**Strengths:**

1. Strong lower bounds are established to illustrate how hard it is to minimize policy regret against the adaptive adversary in the Markov game setting, which is fundamentally different from the bandit learning setting.

2. Efficient algorithms are presented in the paper with strong theoretical guarantees.

**Weaknesses:**

1. For the general $m$-memory bounded adversary, the algorithm developed in the paper requires prior knowledge of $m$.

**Questions:**

1. If the adversary is $m$-memory bounded, could we regard the system state as the combination of $(s_t, \pi_t, \dots, \pi_{t-m+1})$ and then reduce everything to the $0$-memory bounded case?

**Limitations:**

As discussed in the weakness part.

---

> ### Author Rebuttal · Authors · 2024-08-05
>
> We thank the reviewer for the valuable feedback.
>
> ---
> > If the adversary is $m$-memory bounded, could we regard the system state as the combination of $(s_t, \pi_t, \dots, \pi_{t-m+1})$ and then reduce everything to the $0$-memory bounded case?
>
> Thank you for your interesting suggestion. If we augment the past policies as states, the state space will be as large as $S |\Pi|^m$. Since our bound for 1-memory bounded adversary scales polynomially with the number of states, a simple reduction from general $m$ to $m=1$ will not be sample-efficient.
>
> ---
>
> > the algorithm developed in the paper requires prior knowledge of $m$
>
> While our algorithms do require some knowledge of $m$, we would like to clarify that it does not need an exact knowledge of $m$. Any upper bound on $m$ is sufficient for our algorithms.

---

### Decision · Program_Chairs · 2024-09-25

**Decision:**

Accept (poster)

**Comment:**

The paper studies learning in Markov games (MG) in which there is an adversary that is allowed to be adaptive. Authors focus on the policy regret rather than external regret commonly used by many existing work. It is first shown that sublinear policy regret cannot be obtained for the learner. The paper then characterizes the fundamental barriers for the learner to achieve the sublinear regret. The reviewers were positive or slightly positive about this work and the AC agrees with their evaluation.